# CAS-Spec: Cascade Adaptive Self-Speculative Decoding for On-the-Fly Lossless Inference Acceleration of LLMs

**Zhiyuan Ning**[1,2*] **Jiawei Shao**[1] **Ruge Xu**[2] **Xinfei Guo**[2]
**Jun Zhang**[3] **Chi Zhang**[1†] **Xuelong Li**[1†]

[1] TeleAI, [2] Shanghai Jiao Tong University, [3] Hong Kong University of Science and Technology,
{telegraphpolehead,schrodinger,xinfei.guo}@sjtu.edu.cn,
{shaojw2,zhangc120}@chinatelecom.cn, eejzhang@ust.hk, xuelong_li@ieee.org

## Abstract

Speculative decoding has become a widely adopted as an effective technique for lossless inference acceleration when deploying large language models (LLMs). While on-the-fly self-speculative methods offer seamless integration and broad utility, they often fall short of the speed gains achieved by methods relying on specialized training. Cascading a hierarchy of draft models promises further acceleration and flexibility, but the high cost of training multiple models has limited its practical application. In this paper, we propose a novel Cascade Adaptive Self-Speculative Decoding (CAS-Spec) method which constructs speculative draft models by leveraging dynamically switchable inference acceleration (DSIA) strategies, including layer sparsity and activation quantization. Furthermore, traditional vertical and horizontal cascade algorithms are inefficient when applied to self-speculative decoding methods. We introduce a Dynamic Tree Cascade (DyTC) algorithm that adaptively routes the multi-level draft models and assigns the draft lengths, based on the heuristics of acceptance rates and latency prediction. Our CAS-Spec method achieves state-of-the-art acceleration compared to existing on-the-fly speculative decoding methods, with an average speedup from $1.1\times$ to $2.3\times$ over autoregressive decoding across various LLMs and datasets. DyTC improves the average speedup by $47\%$ and $48\%$ over cascade-based baseline and tree-based baseline algorithms, respectively. CAS-Spec can be easily integrated into most existing LLMs and holds promising potential for further acceleration as self-speculative decoding techniques continue to evolve.

## 1 Introduction

Large Language Models (LLMs), such as Llama [1], Mixtral [2], and Qwen [3] are rapidly evolving and increasingly adopted in various applications. However, their autoregressive generation process, where tokens are produced sequentially, coupled with their massive parameter sizes, leads to substantial inference latency and high computational cost. To mitigate this bottleneck, speculative decoding [4–6] has emerged as a highly effective and widely adopted approach, achieving significant speedup without compromising the quality of the generated text. It operates by utilizing a smaller, faster "draft" model to generate a sequence of multiple future tokens concurrently, which are then verified by the larger, more accurate "target" model in parallel, thereby reducing the number of expensive forward passes.

---

*Work done during an internship at TeleAI
†Corresponding Author.

Despite its promise, standard speculative decoding introduces a new burden: it requires training and maintenance of a separate draft model. This not only demands additional training data and computation resources, but also requires careful tuning to ensure compatibility and efficiency aligned with the target model. To address these challenges, self-speculative decoding approaches, self-speculative decoding methods [7–9] have been proposed. These techniques cleverly derive draft predictions from the target model itself, typically by skipping certain blocks in the modules or leveraging model compression techniques, thus eliminating the external training burden. While self-speculation simplifies the deployment pipeline, it often provides limited acceleration. In contrast, cascade speculative decoding [10] introduces a hierarchy of draft models, enabling a multi-stage approach with higher potential speedups and greater flexibility. Yet it requires multiple distinct draft models, making it largely impractical for widespread adoption in real-world systems.

In this paper, we aim to harness the potential of cascade speculation without the crippling overhead of training multiple draft models. We introduce **Cascade Adaptive Self-Speculative Decoding (CAS-Spec)**, a novel framework that brings the performance benefits of cascade speculative decoding without the overhead of training multiple draft models. CAS-Spec dynamically constructs a hierarchy of speculative draft stages using the target model itself by leveraging **Dynamically Switchable Inference Acceleration (DSIA)** strategies. These include techniques such as layer sparsity and activation quantization, enabling the creation of multiple draft models embedded within the target model's inference process. To coordinate this hierarchy at runtime, we further introduce **Dynamic Tree Cascade (DyTC)** algorithm to adaptively route the draft models, construct draft trees as well as controlling the draft lengths. It leverages heuristics based on token acceptance rates and latency prediction to maximize throughput.

Our contributions are summarized as follows:

- We propose CAS-Spec, a novel speculative decoding framework that creates multiple on-the-fly draft stages from a single target model. CAS-Spec achieves lossless inference acceleration without requiring additional draft model training.
- We introduce Dynamic Tree Cascade (DyTC), an adaptive routing algorithm that dynamically manages the draft models and their lengths based on heuristics of acceptance rates and latency predictions. DyTC provides $47\%$ and $48\%$ improvement in average speedup over the cascade-based and tree-based baseline, respectively.
- We demonstrate through extensive evaluations that CAS-Spec achieves state-of-the-art (SOTA) acceleration among on-the-fly speculative decoding methods, delivering speedups ranging from $1.1\times$ to $2.3\times$ over autoregressive decoding across various LLMs and datasets.

This work presents a practical and efficient approach to significantly accelerate LLM inference, paving the way for wider deployment of powerful language models in latency-sensitive and resource-constrained scenarios.

## 2 Preliminary

Autoregressive decoding, where each token depends on the previously generated ones, requires sequential execution, limiting the inference speeds of LLMs. Speculative decoding [4–6] offers a general framework to mitigate this issue by predicting multiple future tokens using a faster draft model $\mathcal{M}_d$ and verifying them with the target model $\mathcal{M}_t$ in parallel. To address practical challenges and unlock greater acceleration, various methods have been developed focusing on the nature and utilization of the draft model, leading to approaches like Self-Speculative Decoding and Cascade Speculative Decoding.

**Self-Speculative Decoding.** Self-speculative decoding (SSD) aims to eliminate the need for external draft models altogether, thereby reducing the overhead of training separate draft models. A common approach of SSD is to derive draft token predictions directly from the target model $\mathcal{M}_t$ itself during the inference process. Several strategies have been explored, including *layer sparsity* [7, 9, 11], *early-exiting* [8, 12, 13], *efficient attention* [14, 15], *Jacobi decoding* [16–18] and *activation quantization* [19]. Other methods such as EAGLE [20–22] and Medusa [23] reuse the target model's hidden states to generate draft tokens efficiently. By leveraging parts of the target model's own information or computation, SSD avoids the burden of separate draft model training and extra memory footprint for maintaining the key-value (KV) cache of the draft LLM.

**Cascade Speculative Decoding.** Vanilla draft models are commonly autoregressive LLMs, which means they can also be accelerated by speculative decoding. Further acceleration is promised by employing multiple draft models, typically ordered by decreasing sizes and latency, e.g., $\{\mathcal{M}_{d_1}, \mathcal{M}_{d_2}, ..., \mathcal{M}_{d_n}\}$. Applying this technique recursively leads to similar approaches known as *vertical cascade* [10], *hierarchical speculative decoding* [15] or *multi-level speculative decoding* [24]. In the direction of draft token generation, since rejecting a draft token means rejecting all the following draft tokens, the acceptance of early draft tokens is more important than the later ones. Cascade Speculative Drafting (CS-Drafting) [10] further proposes *horizontal cascade* which generates early draft tokens using a slightly slower draft model with a high acceptance rate, and subsequent draft tokens with progressively faster draft models. By leveraging *vertical cascade* and *horizontal cascade*, CS-Drafting achieves further acceleration over the vanilla speculative decoding.

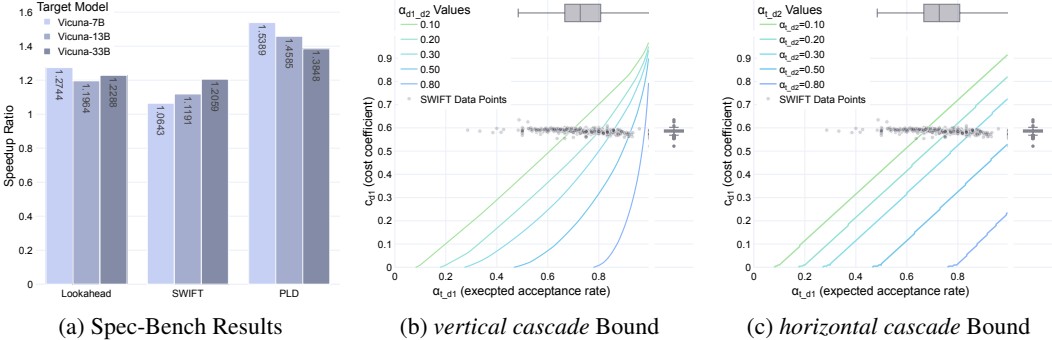

(a) Spec-Bench Results    (b) *vertical cascade* Bound    (c) *horizontal cascade* Bound

Figure 1: (a) Comparison of on-the-fly SSD methods (Lookahead, SWIFT) and methods with statistical draft models (e.g. PLD) on Spec-Bench, tested on NVIDIA H100 GPU. (b) Theoretical effective bound of *vertical cascade* for a draft model $\mathcal{M}_{d_1}$ to be beneficial in the cascade speculative decoding compared with vanilla speculative decoding of $\mathcal{M}_{d_2}$ alone. The x-axis is the expected acceptance rate $\alpha(\mathcal{M}_t, \mathcal{M}_{d_1})$ and the y-axis is the cost coefficient $c(\mathcal{M}_t, \mathcal{M}_{d_1})$. The SWIFT data points are from the Spec-Bench results for `Vicuna-7B-v1.3` model. (The acceptance rates of PLD are between 0.1 and 0.5 in this setting.) (c) Theoretical effective bound of *horizontal cascade*, similar to (b). In this case, we consider $\alpha_{t,d2}$, which is commonly similar to $\alpha_{t,d2}$ in practice.

## 3 Motivation and Analysis

Although self-speculative decoding methods like EAGLE [20–22], BiTA [18] and LayerSkip [8] have shown promising speedups, these methods still require training for several days on a node of 8 common server GPUs like Nvidia A100. While the on-the-fly SSD methods like Lookahead [16] and SWIFT [9] are training-free, they are superior in terms of speed, even falling short of minimal retrieval-based drafting methods like prompt lookup (PLD) [25] on Spec-Bench [26], as shown in Fig. 1a. Since retrieval-based methods like PLD utilize the repeating tokens in the generation process to produce draft tokens, they are inherently lightweight and universally applicable, which weakens the competitiveness of the current training-free SSD methods. To break this trade-off between speed and ease of use, using a combination of multiple training-free SSD methods stands out as a potential solution. Chen et al. [10] present provable improvements over the standard SSD methods by cascading multiple draft models horizontally and vertically. However, not all LLMs have a series of smaller draft models available like the FLAN-T5 [27] family used in CS-Drafting. Requiring multiple draft models is a significant limitation for the practical application of cascade speculative decoding (CSD). We note that the training-free SSD methods can be used to construct a cascade of draft models for speculative decoding. Though it has been proven that a retrieval-based statistical model with negligible cost (e.g. PLD) can almost always gain further speedup against the standard speculative decoding [10], it is not guaranteed that such CSD can be faster than SD with the statistical model alone. It thus leads to the first research question to be addressed:

*RQ 1: Can existing training-free self-speculative decoding methods be used to construct an effective cascade of draft models above retrieval-based methods for on-the-fly speculative decoding?*

Firstly, we establish a theoretical effective bound for an *intermediate draft model* to bring a speedup in CSD. We adopt the same notations as in CS-Drafting for the sake of clarity:

**Expected acceptance rate** $\alpha(\mathcal{M}_t, \mathcal{M}_d)$: The probability that a draft token from a draft model $\mathcal{M}_d$ is verified as correct by the target model $\mathcal{M}_t$.

**Cost coefficient** $c(\mathcal{M}_t, \mathcal{M}_d)$: We define $c(\mathcal{M}_t, \mathcal{M}_d)$ as the ratio of inference times, indicating how much faster the draft model $\mathcal{M}_d$ is compared to the target model $\mathcal{M}_t$ for a single forward step.

**Expected walltime improvement factor (EWIF)** $T$: It represents the predicted gain in overall execution speed (wall time) assuming that the acceptance of each token is an i.i.d. Bernoulli trial.

As proved by Chen et al. [10], the EWIF of speculative decoding is: $T_{SD} = \frac{\phi'_{(\alpha,k)}(1)}{(ck+1)} = \frac{1-\alpha^{k+1}}{(1-\alpha)(ck+1)}$, where $\phi_{(\alpha,k)}(x) = 1 + (x-1)\frac{1-\alpha^{k+1}x^{k+1}}{(1-\alpha x)}$ is the probability generating function of the probability of generating $i$ tokens $p_i$ and $\alpha = \alpha(\mathcal{M}_t, \mathcal{M}_d)$. The EWIF for a *vertical cascade* with two draft models, $\mathcal{M}_{d_1}$ (slower, higher quality) and $\mathcal{M}_{d_2}$ (faster, lower quality), is [10]:

$$T_{VC(\mathcal{M}_{d_1}, \mathcal{M}_{d_2})} = \frac{1 - \alpha\phi^n(\alpha)}{(1-\alpha)(1 + nc_{d_1} + nkc_{d_2})}, \tag{1}$$

where $\phi(x) = \phi_{(\alpha(\mathcal{M}_{d_1}, \mathcal{M}_{d_2}), k)}(x)$, $\alpha = \alpha(\mathcal{M}_t, \mathcal{M}_d)$, and $c_{d_1}, c_{d_2}$ are $c(\mathcal{M}_t, \mathcal{M}_{d_1}), c(\mathcal{M}_t, \mathcal{M}_{d_2})$ respectively. Adapting from Chen et al. [10], Thm. 4.5, the EWIF of *horizontal cascade* for two draft models is:

$$T_{HC(\mathcal{M}_{d_1}, \mathcal{M}_{d_2})} = \frac{\frac{1-\alpha_{d_1}^{k_{d_1}+1}}{1-\alpha_{d_1}} + \alpha_{d_1}^{k_{d_1}}\frac{\alpha_{d_2}(1-\alpha_{d_2}^{k_{d_2}})}{1-\alpha_{d_2}}}{1 + k_{d_1}c_{d_1} + k_{d_2}c_{d_2}}, \tag{2}$$

where $\alpha_{d_1} = \alpha(\mathcal{M}_t, \mathcal{M}_{d_1})$, $\alpha_{d_2} = \alpha(\mathcal{M}_t, \mathcal{M}_{d_2})$, $c_{d_1} = c(\mathcal{M}_t, \mathcal{M}_{d_1})$, $c_{d_2} = c(\mathcal{M}_t, \mathcal{M}_{d_2})$.

Then we get a theoretical bound by solving the inequalities $T_{VC(\mathcal{M}_{d_1}, \mathcal{M}_{d_2})} \geq T_{SD(\mathcal{M}_{d_2})}$ and $T_{HC(\mathcal{M}_{d_1}, \mathcal{M}_{d_2})} \geq T_{SD(\mathcal{M}_{d_2})}$, where $T_{SD(\mathcal{M}_{d_2})} = \frac{1-\alpha_{d_2}^{k_0+1}}{(1-\alpha_{d_2})(c_{d_2}k_0+1)}$. The solution of these inequalities are presented in the Appendix B.

However, the solutions of the inequalities are highly dependent on the hyperparameters of the speculative decoding scheduling, such as $k_{d_1}$, $k_{d_2}$, and $n$.

Thus, we should compare the EWIF of these methods with optimal hyperparameters to find a tighter bound, i.e.

$$\max_{n,k} T_{VC(\mathcal{M}_{d_1}, \mathcal{M}_{d_2})} \geq \max_{k_0} T_{SD(\mathcal{M}_{d_2})}, \quad \max_{k_{d_1}, k_{d_2}} T_{HC(\mathcal{M}_{d_1}, \mathcal{M}_{d_2})} \geq \max_{k_0} T_{SD(\mathcal{M}_{d_2})} \tag{3}$$

This inequality (Eq. 3) does not readily yield a closed-form expression for $c_{d_1}$, as the maximization over integer hyperparameters $(k_0, n, k, k_{d1}, k_{d2})$ typically requires numerical evaluation of given model parameters like the expected acceptance rates and cost coefficients. Therefore we conduct a numerical simulation to find the theoretical effective bound for $\mathcal{M}_{d_1}$ to be benificial in the CSD. When $\mathcal{M}_{d_2}$ is a retrieval-based statistical model with negligible cost, we assume $c_{d_2} = 0.01$ and $\alpha(\mathcal{M}_t, \mathcal{M}_{d_2}) = \alpha(\mathcal{M}_{d_1}, \mathcal{M}_{d_2})$. The simulation results of the borderline of $c_{d_1}$ and $\alpha(\mathcal{M}_t, \mathcal{M}_{d_1})$ are shown in Fig. 1b and Fig. 1c.

With the theoretical effective bound established, we can analyze the performance of existing training-free SSD methods in the context of CSD. For instance, SWIFT [9] employs a layer sparsity strategy, the distribution of its acceptance rates and cost coefficients on Spec-Bench are illustrated in Fig. 1b and Fig. 1c. As observed, most of the data points of SWIFT lie above the theoretical effective bound, indicating that naive HC or VC cascade with SWIFT as the intermediate draft model does not guarantee a speedup over using PLD alone for speculative decoding.

While it's feasible to cascade multiple training-free SSD methods for CSD, more effective cascade algorithms are necessary to fully leverage their potential, with tree-based structures being a promising direction. Since the EWIF of CSD depends on the scheduling algorithm of different draft models, it is possible to achieve a better speedup by adaptively routing the draft models and assigning the draft lengths. This leads to our second research question:

*RQ 2: Can we achieve further speedup by adaptively routing the draft models and assigning the draft lengths, with regards to the characteristics of different DSIA strategies?*

This question is explored in Section 4.2, where we introduce the Dynamic Tree Cascade (DyTC) algorithm for online scheduling of CAS-Spec.

# 4 Cascade Adaptive Self-Speculative Decoding

To address the outlined challenges and leverage the potential of training-free SSD methods, we propose Cascade Adaptive Self-Speculative Decoding (CAS-Spec). CAS-Spec constructs a hierarchy of draft models using various inference acceleration strategies applied to the target model $\mathcal{M}_t$. It then employs a dynamic mechanism to route through this hierarchy and determine draft lengths.

## 4.1 Construct Draft Models with Dynamically Switchable Inference Acceleration Strategies

**Definition 4.1.** A **Dynamically Switchable Inference Acceleration** (DSIA) strategy is a technique that modifies the inference process of a model to accelerate the token generation, which can be dynamically switched on or off during inference. These strategies can often be parameterized (e.g., by the degree of sparsity, quantization bit-width).

Each DSIA strategy, potentially with different parameter settings, can be viewed as creating a distinct "virtual" draft model $\mathcal{M}_{d_i}$ derived from $\mathcal{M}_t$. Examples of DSIA strategies include:

- *Layer Sparsity*: Skipping a subset of transformer layers or a subset of attention and FFN blocks in $\mathcal{M}_t$ to generate draft tokens [7, 9, 11].
- *Early-Exiting*: Using predictions from intermediate layers of $\mathcal{M}_t$ as draft tokens [8, 12, 13].
- *Activation Sparsity*: Keeping only a subset of neurons (activations) in each layer to reduce computation and memory movement of weights. It commonly requires batch size to be 1 or small. [3]
- *Activation Quantization*: Using lower precision (e.g., INT4) for activations and (partial) KV cache of the model during draft generation, as explored by QSpec. It requires a weight-only quantized target model for considerable speedup.
- *Efficient Attention*: Using an efficient attention mechanism like StreamingLLM [30] for draft generation. This is explored in the context of SSD by TriForce and MagicDec. Such methods are commonly more performant in long context generation.

To construct a hierarchy of draft models, there are three approaches in general:

- **Mixing-DSIA Cascade**: Using orthogonal DSIA strategies to create a series of draft models. For example, $\mathcal{M}_{d_1}$ could be a layer-sparse model, while $\mathcal{M}_{d_2}$ could have both layer sparsity and activation sparsity.
- **Replacing-DSIA Cascade**: Using conflicted DSIA strategies to create a series of draft models. For example, $\mathcal{M}_{d_1}$ could be a model with FP8 quantized SageAttention2 [31], and $\mathcal{M}_{d_2}$ could be a model with StreamingLLM attention.
- **Scaling-DSIA Cascade**: Using the same DSIA strategy with different parameter settings (e.g., different degrees of sparsity) to create a series of draft models $\{\mathcal{M}_{d_1}, \mathcal{M}_{d_2}, \dots, \mathcal{M}_{d_n}\}$.

In the spectrum of the trade-off between speed and accuracy, each intermediate draft model $\mathcal{M}_{d_i}$ typically satisfies $\alpha(\mathcal{M}_t, \mathcal{M}_{d_i}) \geq \alpha(\mathcal{M}_t, \mathcal{M}_{d_{i+1}})$ and $c(\mathcal{M}_t, \mathcal{M}_{d_i}) \leq c(\mathcal{M}_t, \mathcal{M}_{d_{i+1}})$. The last model in the hierarchy is expected to be the fastest and least accurate, such as an extremely fast, often non-neural or retrieval-based method.

**Definition 4.2.** In a hierarchy fo draft models $\{\mathcal{M}_{d_1}, \mathcal{M}_{d_2}, \dots, \mathcal{M}_{d_n}\}$, the *bottom draft model $\mathcal{M}_{d_n}$* is a model that serves as the final stage in a cascade of draft models, which means it cannot be further accelerated by speculative decoding.

Prompt Lookup Decoding (PLD) [25] is a prime example. Statistical n-gram models or small, fixed draft heads like those in EAGLE [20–22] and Medusa [23] could also serve this role. However, since EAGLE and Medusa require the hidden states of the target model, their performance will be greatly affected when using the hidden states from DSIA draft models. For simplicity and considering its proven efficacy in CS-Drafting [10], we often consider PLD as a default $\mathcal{M}_{d_n}$.

The $\mathcal{M}_{d_i}$ are inherently training-free if the DSIA strategies are training-free. By choosing a training-free bottom draft model and DSIA strategies, CAS-Spec can be implemented without training multiple draft models and thus more easily integrated into a wide range of LLMs. Among the listed DSIA strategies, layer sparsity does not require special condition to gain speedup. We choose it for easy

---

[3]There are no existing works on SSD with activation sparsity yet, but it is a promising direction with recent advances in this technique [28, 29]

comparison with other speculative decoding methods. CAS-Spec with other DSIA strategies like activation sparsity and quantization is discussed in the Appendix C.

## 4.2 Dynamic Tree Cascade (DyTC)

Tree attention allows for more flexible and efficient speculative decoding by enabling the parallel verification of different branches of draft tokens [23, 32]. Similarly, early tokens in a draft token tree should also be prioritized for more accepted tokens.

**Proposition 4.3.** *Tree Cascade (TC) assigns the different draft models in the draft token tree to maximize the expected acceptance rate of the early draft tokens.*

Given a set of available DSIA strategies and a bottom draft model, there are many possible configurations for the cascade of draft models, forming a set of candidate draft models.

In CAS-Spec systems, the challenge is to determine *where* to start generating tokens (in the draft token tree), *which* draft models to use, and *when* to switch between them. Since the dimensions of the search space are large, it is impractical to use global optimization methods, which are commonly used in vanilla speculative decoding to find the optimal draft length. Instead, we propose to use a heuristic-based approach to dynamically adapt the hyperparameters of the scheduling algorithm during inference. This dynamic adaptation is guided by heuristics based on continuously updated estimates of acceptance rates and latency predictions. Inspired by the idea of dynamic draft tree expansion [33], we propose to leverage online optimization to adapt the expansion of *Tree Cascade*:

**Proposition 4.4.** *Dynamic Tree Cascade (DyTC) is a dynamic scheduling algorithm that adaptively selects the draft models and their configurations in a tree structure based on the online acceptance rates and latency predictions.*

**Acceptance Rate for Draft Configurations.** For each potential draft model configuration $\mathcal{M}_{d_i}$ (including DSIA variants of $\mathcal{M}_t$ and vertical cascade combinations), DyTC maintains an online estimate $\hat{\alpha}(\mathcal{M}_t, \mathcal{M}_{d_i})$ of its acceptance rate. This estimate is continuously updated using an Exponential Moving Average (EMA) mechanism:

$$\hat{\alpha}_{new} = \lambda \cdot \hat{\alpha}_{prev} + (1 - \lambda) \cdot \hat{\alpha}_{recent} \tag{4}$$

where $\hat{\alpha}_{prev}$ is the estimate from the previous step, $\hat{\alpha}_{recent}$ is the acceptance rate computed from a local history window of the most recent $H$ generation steps (we use $H = 20$ and $\lambda = 0.7$ in our experiments), and $\lambda$ controls the balance between stability and responsiveness to changing generation contexts.

Critically, for computing $\hat{\alpha}_{recent}$, we focus on the acceptance of the *first* draft token generated by each configuration, rather than the overall ratio of accepted to drafted tokens. Specifically, if the local history contains outcomes $\{o_1, o_2, \ldots, o_H\}$ where $o_i \in \{0, 1\}$ indicates whether the first drafted token was accepted, then: $\hat{\alpha}_{recent} = \frac{1}{H} \sum_{i=1}^{H} o_i$.

The EMA-based update ensures that $\hat{\alpha}$ adapts to the dynamic nature of text generation, where acceptance probability can vary significantly across different tasks (e.g., translation vs. summarization) and even within a single sequence. For cold starts initialization, a brief calibration phase can be performed at the beginning of generation to gather initial acceptance statistics for each configuration, detailed in Appendix D.

**Token-Level Information for Drafted Tokens.** While the configuration selection in `FindBestConfigurationForStep` (Algorithm 2) relies on the aggregate acceptance rate estimates described above, the calculation of *accumulated acceptance rate* $\prod_{j=1}^{l_s} \hat{\alpha}_j$ for already-drafted but not-yet-verified tokens *does* incorporate token-level information. Specifically, we consider:

- For neural draft models: the normalized probability (logit) of the drafted token, which is positively correlated with acceptance likelihood [33].
- For non-neural drafts (e.g., PLD): longer length of the n-gram match indicating higher confidence.

This token-level refinement allows DyTC to more accurately estimate the quality of different candidate branches in the draft tree when selecting the next leaf node to expand. However, for *future* tokens (those not yet generated), we cannot access such token-level information without actually running the draft model, which would be prohibitively expensive for all candidate configurations. Therefore, the configuration selection relies on the historical acceptance rate estimates.

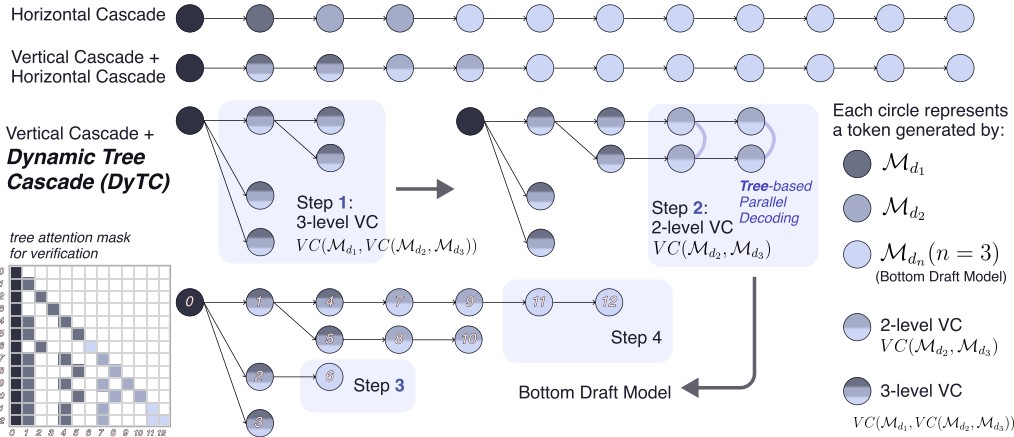

Figure 2: Illustration a example of the Dynamic Tree Cascade (DyTC) algorithm when $n = 3$.

**Hardware-Aware Latency Prediction** ($\hat{c}$): The cost coefficient $c(\mathcal{M}_t, \mathcal{M}_{d_i})$ depends on the specific DSIA strategy and the hardware platform. DyTC utilizes a latency model to predict these costs. We predicted the roofline latency of the hardware platform with Bayesian linear regression.

**DyTC Algorithm.** Firstly, we start by generating tokens at the leaf node where the *accumulated acceptance rate* $\prod_{j=1}^{l} \hat{\alpha}_j$ is the highest. Here $l$ is the path length from the root node to the leaf node and $\hat{\alpha}_j$ is the estimated acceptance rate of the $j$-th node in the path. At each decoding step, DyTC evaluates different cascade configurations. A "configuration" involves a selection of DSIA models, their draft lengths, and their arrangement. For simplicity, we consider a round of *vertical cascade* as a holistic step. For example, at the $s$-th step of generation, DyTC considers:

- Generate with a single model $\mathcal{M}_{d_i}$, with a draft length of $k_s$ (commonly small for finer control).
- A *vertical cascade*: $VC(\mathcal{M}_{d_i}, (\dots, \mathcal{M}_{d_n}))$, with an *expected* draft length of $k_s$ (the draft length cannot be strictly controlled since it depends on the acceptance of the low-level draft tokens).
- End the generation of the draft tokens (if the *accumulated acceptance rate* $\prod_{j=1}^{l_s} \hat{\alpha}_j$ is too low).

The decision-making process aims to maximize the overall EWIF, choosing proper configurations $\mathcal{M}_{d_s}$ and $k_s$. Firstly, we start with a *Greedy Search* that optimize the predicted local speedup $\hat{t_s} = \frac{\hat{\alpha}(1-\hat{\alpha}^{k_s})}{(1-\hat{\alpha})\hat{c}k_s} \prod_{j=1}^{l_s} \hat{\alpha}_j$ of the current step. However, the speedup of CSD may not obey the Greedy Choice Property, that is, choosing the locally optimal solution at each step does not guarantee a globally optimal EWIF. For instance, if there are two draft models: $\mathcal{M}_{d_1}$ with $\hat{\alpha_{d_1}} = 0.9$ and $\hat{c_{d_1}} = 0.4$, and $\mathcal{M}_{d_2}$ with $\hat{\alpha_{d_2}} = 0.8$ and $\hat{c_{d_2}} = 0.3$, the local speedup when $k_s = 1$ is $\hat{t_s}(\mathcal{M}_{d_1}) = \frac{0.9}{0.4} \approx 2.25$ and $\hat{t_s}(\mathcal{M}_{d_2}) = \frac{0.8}{0.3} \approx 2.67$ at the first step. The *Greedy Search* algorithm would select $\mathcal{M}_{d_2}$ at every step, with a suboptimal overall EWIF of $1.554$. On the other hand, if we use the *horizontal cascade* of $\mathcal{M}_{d_1}$ and $\mathcal{M}_{d_2}$, we can achieve an overall EWIF of $1.615$.

Dynamic programming could be used to find the optimal global solution, but the computational overhead is prohibitive for online scheduling since the search space grows exponentially with the number of steps. Inspired by the concept of "admissible heuristic" from the A* algorithm[34], we propose to adjust the local optimization target considering not only the estimated speedup brought by the current step, but also the estimated *least future speedup* to address this issue. We define the *least future speedup* as the EWIF of using the Bottom Draft model $\mathcal{M}_{d_n}$ for the following draft step. So the subproblem of the $s$-th step is to maximize the following objective function:

$$\mathcal{T}_s(\mathcal{M}_{d_s}, k_s) \prod_{j=1}^{l_s} \hat{\alpha}_j, \quad \text{where } \mathcal{T}_s(\mathcal{M}_{d_s}, k_s) = \frac{\frac{\hat{\alpha}(1-\hat{\alpha}^{k_s})}{1-\hat{\alpha}} + \hat{\alpha}^{k_s}\alpha_{\hat{d}_n}}{\hat{c}k_s + c_{\hat{d}_n}} \tag{5}$$

Since $\prod_{j=1}^{l_s} \hat{\alpha}_j$ only depends on the chosen leaf node, we first find the best leaf node with the highest *accumulated acceptance rate*. We stop the generation of the draft tokens if $\frac{\alpha_{\hat{d}_n}}{c_{\hat{d}_n}} \prod_{j=1}^{l_s} \hat{\alpha}_j < t_{min}$,

**Algorithm 1:** Dynamic Tree Cascade (DyTC) Draft Generation

---

**Input** : Initial prefix $x_0$, Maximum tree size $M_{tree\_max}$
Set of candidate draft model configurations $\mathcal{S}$, Bottom draft model $\mathcal{M}_{d_n}$ with $\hat{\alpha}_{d_n}, \hat{c}_{d_n}$
Minimum overall speedup threshold $t_{min}$, Maximum draft length per expansion step $k_{max}$
Top-K token selection $K$, Top-P tree probability threshold $P_{tree}$
**Output :** Generated draft token tree $T_r$

---

1 Initialize $T_r$ with $N_{root}$ representing the last bonus token $x_0$
2 Dictionary tracking accumulated acceptance rate $P_{acc}[N_{root}] \leftarrow 1.0$
3 Dictionary tracking active nodes $D_{active}[N_{root}] \leftarrow True$  // Mark $N_{root}$ as active leaf
4 **while** $T_r.size() < M_{tree\_max}$ **do**
5     $N_{leaf} \leftarrow arg\max_{N \in T_r}\{P_{acc}[N] \mid D_{active}[N] = True\}$
6     **if** $N_{leaf}$ *is* null **then**
7       **break**                          // No more active leaves to expand
8     $(S^*, k^*) \leftarrow$ FindBestConfigurationForStep$(\mathcal{S}, \hat{\alpha}_{d_n}, \hat{c}_{d_n}, k_{max})$
9     **if** $S^*$ *is* null **then**
10       $D_{active}[N_{leaf}] \leftarrow False$
11       **break**                    // No beneficial configuration, stop expansion
12     $x \leftarrow$ GetSequenceToNode$(N_{leaf})$
13     $x_{siblings} \leftarrow$ GetSiblingTokens$(N_{leaf})$
14     **if** $S^*$ *is not* $\mathcal{M}_{d_n}$ ***And*** $x_{siblings}$ *is not empty* **then**
15       $x \leftarrow$ concat$(x, x_{siblings})$          // Tree-based sequence parallelism
16     $\mathbf{y} \in \mathbb{N}^{k^* \times K} \leftarrow$ GenerateDraftTokens$(S^*, x, k^*)$
17     $\hat{\alpha}_{S^*} \leftarrow$ current estimate $\hat{\alpha}(S^*)$
18     **for** $i \leftarrow 1$ **to** $k^*$ **do**
19       **for** $j$ *in* $arg\,top_P\mathbf{y}[i,:]$ **do**
20         $N_{parent} \leftarrow$ GetSiblingNode$(N_{leaf}, j)$       // returns $N_{leaf}$ if $j = 0$
21         $N_{new} \leftarrow T_r.$add_child$(N_{parent}, \mathbf{y}[i,j],$ info from $S^*)$
22         **if** $j == 0$ **then**
23           $N_{first} \leftarrow N_{new}$
24         $P_{acc}(N_{new}) \leftarrow P_{acc}(N_{parent}) \times \hat{\alpha}_{S^*}$
25         $D_{active}[N_{new}] \leftarrow True$
26       $N_{parent} \leftarrow N_{first}$
27       **if** $T_r.size() \geq M_{tree\_max}$ **then**
28         **return** $T_r$                  // Tree size limit reached
29 **return** $T_r$

---

where $t_{min}$ is a threshold for the minimum local speedup, or the total tree size exceeds the maximum size $m$. Then we get the best configuration $\mathcal{M}_{d_s}$ and $k_s$ by solving the optimization problem in Eq. 5:

$$\mathcal{M}_{d_s}, k_s = \arg\max_{\mathcal{M}_{d_s}, k_s} \mathcal{T}_s(\mathcal{M}_{d_s}, k_s), \quad \text{s.t. } k_s \in [1, k_{\max}] \tag{6}$$

**Tree-based Parallel Draft Generation.** Draft models constructed with DSIA strategies are themselves an LLM variant, with memory-bounded inference process. Tree attention allows for not only parallel verification of multiple candidate draft paths, but also parallel generation of draft paths. Following the idea of tree-based parallel decoding in SpecInfer [32], we can generate draft tokens for multiple sibling leaf nodes in parallel. Given the TOP-K selected siblings $N_{s_1}, N_{s_2}, \ldots, N_{s_m}$ ($m = K - 1$) of the selected leaf node $N_{leaf}$, we select TOP-P sibling nodes based on the normalized probability of the drafted candidate tokens. In memory-bounded decoding process, a slightly larger sequence of input tokens each step is acceptable and does not significantly affect the overall latency of draft generation. Thus, we use the same draft length $k_s$ for all selected sibling nodes to simplify the implementation.

The algorithm is summarized in Alg. 1. The detailed algorithms for functions FindBestLeafNode, FindBestConfig, and GenerateDraftToks are presented in Appendix D.

Table 1: Overall speedup compared to Autoregressive Decoding on Spec-Bench. Models: `Vicuna-7B-v1.3`, `Vicuna-13B-v1.3`, and `Vicuna-33B-v1.3`. [36] **Bold** indicates the best performance among training-free methods. Underlined indicates the best overall performance, including methods with training. *CAS-Spec*[†] denotes CAS-Spec with Kangaroo and PLD

| Model | Method | MT-Bench | Trans | Summary | QA | Math | RAG | Overall |
|-------|--------|----------|-------|---------|-----|------|-----|---------|
| **7B** | Lade | 1.386 | 1.172 | 1.173 | 1.253 | 1.567 | 1.078 | 1.274 |
| | PLD | 1.563 | 1.046 | **2.276** | 1.109 | 1.603 | 1.642 | 1.539 |
| | SWIFT | 1.073 | 1.075 | 1.096 | 1.019 | 1.067 | 1.055 | 1.064 |
| | **CAS-Spec** | **1.598** | **1.103** | 2.268 | **1.145** | **1.664** | **1.676** | **1.578** |
| | *Kangaroo* | 1.698 | 1.307 | 1.548 | 1.409 | 1.658 | 1.581 | 1.534 |
| | ***CAS-Spec**[†]* | 1.727 | 1.312 | 2.327 | 1.407 | 1.701 | 1.695 | 1.696 |
| **13B** | Lade | 1.281 | 1.065 | 1.132 | 1.128 | 1.480 | 1.068 | 1.196 |
| | PLD | 1.418 | 1.020 | **2.104** | 1.035 | 1.577 | 1.673 | 1.458 |
| | SWIFT | 1.155 | 1.087 | 1.196 | 1.040 | 1.106 | 1.142 | 1.119 |
| | **CAS-Spec** | **1.562** | **1.134** | 2.063 | **1.107** | **1.582** | **1.691** | **1.524** |
| | *Kangaroo* | 1.652 | 1.244 | 1.483 | 1.340 | 1.652 | 1.508 | 1.482 |
| | ***CAS-Spec**[†]* | 1.732 | 1.251 | 2.337 | 1.401 | 1.719 | 1.689 | 1.673 |
| **33B** | Lade | 1.295 | 1.085 | 1.159 | 1.165 | 1.535 | 1.114 | 1.229 |
| | PLD | 1.431 | 1.047 | **1.891** | 1.061 | 1.523 | 1.396 | 1.385 |
| | SWIFT | 1.218 | **1.187** | 1.244 | 1.152 | 1.218 | 1.221 | 1.206 |
| | **CAS-Spec** | **1.547** | 1.176 | 1.862 | **1.186** | **1.563** | **1.490** | **1.481** |

## 5 Experiments

We conduct comprehensive experiments to evaluate the effectiveness of our proposed Cascade Adaptive Self-Speculative Decoding (CAS-Spec) algorithm. We aim to answer the two research questions posed in Section 3.

### 5.1 Experimental Setup

We evaluate CAS-Spec on a range of widely-used open-source LLMs, including `Llama-2-7B` [35] and `Vicuna-v1.3` [36] family. These models represent different architectures and training objectives, allowing us to assess the generalizability of CAS-Spec. All experiments ensure lossless decoding, meaning that the output is identical to that of standard autoregressive decoding. We choose Spec-Bench [26] and for evaluation. Spec-Bench is a comprehensive benchmark including various tasks such as multi-turn conversations, mathematical reasoning, and summarization. For all datasets, we measure the generation speed for producing 1024 new tokens. To demonstrate the versatility of CAS-Spec, we conduct experiments on server-grade GPU (NVIDIA H100 80GB). The primary metric is **Speedup**, defined as the wall time of autoregressive decoding divided by the wall time of the speculative decoding method.

**CAS-Spec Configuration.** For CAS-Spec, we construct a hierarchy of draft models using the following DSIA strategies, chosen for their training-free nature and effectiveness:

- **DSIA (Layer Sparsity):** Skip every other Transformer layer in $\mathcal{M}_t$, following SWIFT [9].
- **DSIA (Early Exiting):** Exit after a subset of Transformer layers, then decode with a trained adapter, following Kangaroo [13]. (Kangaroo is a non-training-free method and only provides the adapter weights for 7B and 13B models.)
- **bottom draft model** ($\mathcal{M}_{d_n}$)**:** Prompt Lookup Decoding (PLD) is used as the final, fastest stage, as it has negligible computational cost.

Our CAS-Spec implementation primarily uses a three-level DSIA cascade: $\mathcal{M}_{d_1}$ and $\mathcal{M}_{d_2}$, followed by $\mathcal{M}_{d_n}$ (PLD). The DyTC algorithm dynamically selects between these options and adjusts draft lengths ($k_{max}$ set to 5, $t_{min}$ set to 1.1). Detailed configurations and hyperparameters for CAS-Spec are provided in Appendix E.

## 5.2 Main Results

Table 1 summarizes the main speedup results. CAS-Spec consistently outperforms all baseline on-the-fly (training-free) speculative decoding methods across all tested models, and datasets. CAS-Spec achieves speedups ranging from $1.10\times$ to $2.27\times$. This significantly surpasses individual training-free methods like PLD and SWIFT. Notably, CAS-Spec's performance is competitive with, and in some cases exceeds, reported numbers for Kangaroo, despite CAS-Spec being entirely training-free. With trained methods like Kangaroo, which leverage a small tuned head for early exiting, we can achieve more substantial gains over both PLD and Kangaroo. The detailed comparison between training-free and not-training-free methods is provided in Appendix F.1. As shown in Figure 3, for `Vicuna-7B-v1.3`, CAS-Spec achieves an average speedup of $47\%$ over CS-Drafting [10](VC+HC) and $48\%$ over tree algorithm in SWIFT [9].

## 5.3 Discussion

The experimental results robustly demonstrate that CAS-Spec achieves SOTA speedups among on-the-fly speculative decoding methods. **Addressing RQ1**: Our findings confirm that training-free self-speculative methods can be effectively layered to construct a cascade that significantly outperforms a single, strong statistical draft model like PLD under proper cascade scheduling. **Addressing RQ2**: The ablation study on DyTC clearly shows its superiority over static cascade scheduling. The ability to dynamically route through the draft model hierarchy and assign draft lengths based on runtime heuristics (acceptance rates and latency predictions) is crucial for maximizing performance. This adaptability allows CAS-Spec to handle variations in generation difficulty and hardware characteristics more effectively, boosting real-world applications [37, 38].

The training-free nature of CAS-Spec, combined with its high performance, makes it a practical and attractive solution for accelerating LLM inference in diverse deployment scenarios. It can be readily integrated with existing LLMs without the need for costly retraining or maintaining separate draft model weights and KV caches (beyond the DSIA modifications to the target model's inference path).

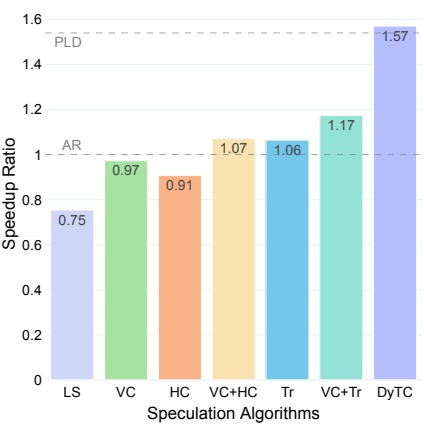

Figure 3: Speedup of different methods relative to baseline. AR (1.0) and PLD (1.54) reference lines are shown. The vertical line separates two groups of methods.

## 6 Conclusion

In this paper, we introduced Cascade Adaptive Self-Speculative Decoding (CAS-Spec), a novel algorithm for lossless LLM inference acceleration that eliminates the need for training separate draft models. CAS-Spec constructs a hierarchy of speculative draft models by leveraging dynamically switchable inference acceleration (DSIA) strategies, applied to the target model. This approach offers significant flexibility and ease of integration.

A core contribution of our work is the Dynamic Tree Cascade (DyTC) method. DyTC adaptively routes generation through the multi-level draft models and assigns draft lengths based on online heuristics of acceptance rates and latency predictions. This dynamic scheduling allows CAS-Spec to optimize its performance continuously during inference.

Our experiments demonstrate that CAS-Spec achieves state-of-the-art acceleration among on-the-fly speculative decoding methods. The results validated our hypotheses that (1) training-free DSIA strategies can form effective cascades over fast bottom draft models, and (2) dynamic scheduling via DyTC further enhances these gains.

CAS-Spec offers a compelling solution for practical LLM deployment by providing substantial speedups without the overheads associated with training and maintaining external draft models. Its adaptability and ease of integration make it a valuable tool for a wide range of LLMs.

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

## A    Limitations and Future Work

While CAS-Spec demonstrates promising performance, its effectiveness is inherently tied to the quality and efficiency of the chosen DSIA strategies. The online estimation for DyTC, though lightweight, introduces some computational overhead and require a short warm-up period for optimal performance. Moreover, since dynamic scheduling depends on the heuristics of the generation of a certain token sequence, its improvement is lessened when the batch size is large. Other limitations include the incompatibility of CAS-Spec with current SOTA speculative decoding methods like EAGLE3 due to their reliance on hidden states of the verify model.

Future research could explore integrating more sophisticated or newly developed training-free SSD techniques as DSIA components within the CAS-Spec framework. Further refinement of the DyTC algorithm, potentially incorporating more advanced online learning for scheduling, could yield additional improvements. Investigating the application of CAS-Spec to even larger models or different modalities, and exploring hardware co-design to optimize DSIA strategy execution, are also promising avenues for future work. As the field of self-speculative decoding continues to evolve, CAS-Spec provides a flexible and powerful framework to leverage these advancements.

## B    Theoretical Analysis

For certain hyperparameters $(k_0, n, k, k_{d1}, k_{d2})$, we can derive the theoretical bounds for the cost coefficient of $\mathcal{M}_{d_1}$, such that the speedup of the cascade is greater than the speedup of standard speculative decoding with $\mathcal{M}_{d_2}$ alone.

For *vertical cascade*, the solution of the inequality $T_{VC(\mathcal{M}_{d_1}, \mathcal{M}_{d_2})} \geq T_{SD(\mathcal{M}_{d_2})}$ gives the bound:

$$c_{d_1} \leq \frac{1}{n} \left[ \left( \frac{1 - \alpha_{d_1}(\phi(\alpha_{d_1}))^n}{1 - \alpha_{d_1}} \right) \left( \frac{(1 - \alpha_{d_2})(c_{d_2} k_0 + 1)}{1 - \alpha_{d_2}^{k_0 + 1}} \right) - (1 + nk c_{d_2}) \right]$$

where $\phi(\alpha_{d_1}) = \frac{1 - \alpha_{d_1}^{k_{d_1} + 1}}{(1 - \alpha_{d_1})(1 + k_{d_1} c_{d_1})}$.

For *horizontal cascade*, the solution of the inequality $T_{HC(\mathcal{M}_{d_1}, \mathcal{M}_{d_2})} \geq T_{SD(\mathcal{M}_{d_2})}$ gives the bound:

$$c_{d_1} \leq \frac{1}{k_{d_1}} \left[ \left( \frac{1 - \alpha_{d_1}^{k_{d_1} + 1}}{1 - \alpha_{d_1}} + \alpha_{d_1}^{k_{d_1}} \frac{\alpha_{d_2}(1 - \alpha_{d_2}^{k_{d_2}})}{1 - \alpha_{d_2}} \right) \left( \frac{(1 - \alpha_{d_2})(c_{d_2} k_{d_2} + 1)}{1 - \alpha_{d_2}^{k_{d_2} + 1}} \right) - (1 + k_{d_2} c_{d_2}) \right]$$

## C    DSIA

Other than the layer sparsity, we also consider the activation quantization and activation sparsity as two candidates of CAS-Spec framework. For activation quantization, we refer to the implementation of QSpec. While for activation sparsity, we adopt the TEAL's method of this DSIA strategy.

Furthermore, since the activation quantization and activation sparsity are both orthogonal to the layer sparsity, we can construct draft models with Mixing-DSIA Cascade. For instance, we can construct $\mathcal{M}_{d_1}$ as the W4A4 quantized model, and $\mathcal{M}_{d_2}$ as the W4A4 model with layer sparsity and activation sparsity.

However, using these two DSIA strategies inherently requires a weight-only quantized draft model with batch size of 1. This configuration is suitable for edge inference, but not general enough for large-scale inference scenarios. Therefore, we do not include the experiments on these two DSIA strategies in our CAS-Spec framework by now.

## D    Dynamic Tree Cascade Algorithm

**Maintaining Estimates for Inactive Configurations.** For draft configurations not currently selected in a given step, their acceptance rate estimates are preserved and do not decay. When a previously unused configuration becomes active again (e.g., due to changing generation context), its estimate is updated based on actual verification outcomes. In cold-start scenarios or for configurations that have

never been used, we initialize estimates using either: (1) brief offline profiling on representative tasks, or (2) heuristic priors based on the DSIA strategy's aggressiveness (e.g., higher layer sparsity implies lower expected acceptance rate).

Note that for vertical cascade configurations like $VC(\mathcal{M}_{d_i}, \mathcal{M}_{d_n})$, we maintain a single acceptance rate estimate corresponding to the highest-level draft model $\mathcal{M}_{d_i}$, since all draft tokens are ultimately verified by this model before proceeding to $\mathcal{M}_t$.

The detailed functions mentioned in the Algorithm 1 are as follows:

---

**Algorithm 2:** Dynamic Tree Cascade (DyTC) Functions

---

**1 Function** FindBestConfigurationForStep($\mathcal{S}_{candidates}, \hat{\alpha}_{d_n}, \hat{c}_{d_n}, k_{max}$)**:**

2    $\mathcal{M}_{best\_choice} \leftarrow$ null; $k_{best\_choice} \leftarrow 0$; $max\_obj\_val \leftarrow -\infty$

3    **foreach** *configuration* $S \in \mathcal{S}_{candidates}$ **do**

4      $\hat{\alpha}_S \leftarrow$ current estimate $\hat{\alpha}(S)$

5      $\hat{c}_S \leftarrow$ current estimate $\hat{c}(S)$

6      **for** $k \leftarrow 1$ **to** $k_{max}$ **do**

7        **if** $\hat{c}_S k + \hat{c}_{d_n} \approx 0$ **then**

8          **continue**

9        float $E_{accepted}$; **if** $\hat{\alpha}_S \approx 1.0$ **then**

10         $E_{accepted} \leftarrow k$

11        **else**

12          $E_{accepted} \leftarrow (\hat{\alpha}_S(1 - \hat{\alpha}_S^k))/(1 - \hat{\alpha}_S)$

13        float $\mathcal{T}_{val} \leftarrow (E_{accepted} + \hat{\alpha}_S^k \hat{\alpha}_{d_n})/(\hat{c}_S k + \hat{c}_{d_n})$

14        **if** $\mathcal{T}_{val} > max\_obj\_val$ **then**

15          $max\_obj\_val \leftarrow \mathcal{T}_{val}$

16          $\mathcal{M}_{best\_choice} \leftarrow S$

17          $k_{best\_choice} \leftarrow k$

18    **if** $max\_obj\_val \leq 0$ **then**

19      **return** (null, $0$)

20    **return** ($\mathcal{M}_{best\_choice}, k_{best\_choice}$)

---

## E   Experimental Details

In this section, we provide the details of our experimental setup, specifically for CAS-Spec configuration in the main experiment. Our draft models are constructed using mainly layer sparsity based on SWIFT, which is a training-free and on-the-fly DSIA strategy. The hierarchy is got by Scaling-DSIA Cascade, which tunes the layer sparsity to get different draft models. We also consider the Prompt Lookup Decoding (PLD) as the bottom draft model. The configuration of CAS-Spec is as follows: $\mathcal{M}_{d_1}$ has around 0.4 layer sparsity, while $\mathcal{M}_{d_2}$ has around 0.6 layer sparsity. Thus it leads to multiple candidate $\mathcal{M}_{d_s}$ for each step:

- basic model: $\mathcal{M}_{d_1}(LS = 0.4)$, $\mathcal{M}_{d_2}(LS = 0.6)$, $\mathcal{M}_{d_3}$ (PLD)
- 2-Level VC: $VC(\mathcal{M}_{d_1}, \mathcal{M}_{d_3})$, $VC(\mathcal{M}_{d_2}, \mathcal{M}_{d_3})$
- 3-Level VC: $VC(\mathcal{M}_{d_1}, VC(\mathcal{M}_{d_2}, \mathcal{M}_{d_3}))$

However, due to the insufficient gap between the layer sparsity of $\mathcal{M}_{d_1}$ and $\mathcal{M}_{d_2}$, the 3-Level VC configuration turns out to be less efficient and hence rarely used in the DyTC algorithm. Therefore, we present the results of 2-Level VC in the main experiment.

## F   Additional Experimental Results

Figure 3 shows the speedup of different methods on `Vicuna-7B-v1.3` model. LS refers to the layer sparsity only drafting without tree attention. `VC,HC,VC+HC` are the framework of using the vertical

cascade and horizontal cascade with PLD, based on the implementation of CS-Drafting. `Tr` is the standard SWIFT implementation with Tree Attention. `Tr+VC` follows the same implementation of CS-Drafting, but with the tree attention. `DyTC` is the version with Dynamic Tree Cascade algorithm. Compared to `VC+HC` configuration, `DyTC` achieves a 73% improvement in average speedup. Compared to `Tr` (SWIFT), `DyTC` achieves a 47% improvement in average speedup.

## F.1 Comparison with Trained Methods

Table 2: Comparison with trained methods on `Vicuna-7B-v1.3` model.

| Method | Training-Free | #Mean accepted tokens | Speedup |
|---|---|---|---|
| PLD | Yes | 1.75 | 1.54x |
| SWIFT | Yes | 3.01 | 1.06x |
| CAS-Spec (SWIFT,PLD) | Yes | 3.43 | 1.58x |
| Speculative Decoding (Vicuna 68m) | No | 2.27 | 1.44x |
| Medusa | No | 2.39 | 1.69x |
| EAGLE | No | 3.57 | 2.05x |
| EAGLE2 | No | 4.36 | 2.21x |

In comparison with trained speculative decoding methods, our CAS-Spec framework with SWIFT and PLD achieves higher speedup than vanilla speculative decoding methods with Vicuna 68m, while being training-free. The gap between training-free and not-training-free methods is significant, as trained methods like Medusa and EAGLE achieve higher acceptance rates and speedups, leveraging the full model's capabilities.

