# OpenReview forum: "CAS-Spec: Cascade Adaptive Self-Speculative Decoding for On-the-Fly Lossless Inference Acceleration of LLMs"
_NeurIPS.cc/2025/Conference — NeurIPS 2025 poster_

### Official Review · Reviewer_7MsQ · 2025-07-02

**Clarity:** 3
**Significance:** 3
**Originality:** 3
**Rating:** 4
**Confidence:** 3

**Summary:**

This paper proposes Cascade Adaptive Self-Speculative Decoding (CAS-Spec), a novel method for accelerating LLM inference without additional training. It constructs speculative draft models using dynamically switchable acceleration techniques such as layer sparsity and activation quantization. To further enhance efficiency, the authors introduce the Dynamic Tree Cascade (DyTC) algorithm, which adaptively routes draft models based on acceptance rates and latency estimates. CAS-Spec delivers incremental improvements over existing training-free baseline methods.

**Questions:**

1. Could the authors provide more details on the computational complexity or runtime overhead associated with applying CAS-Spec?

2. Additionally, in high-throughput scenarios (e.g., with batch sizes greater than 1), does CAS-Spec maintain its speedup advantage compared to baseline methods?

**Ethical Concerns:**

["NO or VERY MINOR ethics concerns only"]

**Final Justification:**

In light of the additional experiments provided by the authors, I will retain my present assessment.

**Limitations:**

Yes.

**Paper Formatting Concerns:**

N/A.

**Quality:**

2

**Strengths And Weaknesses:**

**Strengths**

1. CAS-Spec addresses training-free speculative decoding, offering practical value in scenarios where additional training is impractical or unavailable.

2. The method dynamically integrates existing training-free acceleration techniques to build a more efficient draft model.

3. This paper targets a practical deployment scenario where multiple draft models are not always available for a given LLM.


**Weakness**

The main concern is the relatively modest speedup achieved by CAS-Spec. As shown in Table 1, for 7B-scale LLMs, CAS-Spec outperforms the second-best method (PLD) by only 0.037 on average. The improvements for 13B and 33B models are similarly limited. Given the added complexity of dynamic switching and the DyTC algorithm, the marginal gains makes the overall contribution of CAS-Spec appear somewhat incremental.

A minor issue: Section 3 states that self-speculative decoding methods require substantial training time (e.g., several days or weeks). This appears to overstate the cost—such methods typically use a few billion tokens for training and have relatively few trainable parameters. Training for weeks seems unrealistic. While training-free approaches are valuable, training-based methods can also be efficient and high-performing, and should be evaluated fairly.

---

> ### Author Rebuttal · Authors · 2025-07-31
>
> We sincerely thank the reviewer for their thoughtful evaluation and for recognizing the practical value and novelty of CAS-Spec. We are encouraged by the positive assessment of our work's relevance to real-world deployment scenarios. Your insightful comments have helped us identify areas where we can provide greater clarity.
>
> We address your concerns and questions below.
>
> **[W1]. On the "Modest" Speedup and Incremental Contribution**
> > The main concern is the relatively modest speedup achieved by CAS-Spec. As shown in Table 1, for 7B-scale LLMs, CAS-Spec outperforms the second-best method (PLD) by only 0.037 on average. The improvements for 13B and 33B models are similarly limited. Given the added complexity of dynamic switching and the DyTC algorithm, the marginal gains makes the overall contribution of CAS-Spec appear somewhat incremental.
>
> We appreciate the reviewer’s sharp analysis of the results in Table 1. While the overall average speedup over the second-best method (PLD) may appear modest, it's more difficult get further speedup over PLD than it is over SWIFT. This is limited by the inefficiency of current training-free self-speculative decoding methods like SWIFT when it comes to small models.
> With trained methods like Kangaroo, which leverage a small tuned head for early exiting, we can achieve more substantial gains over both PLD and Kangaroo:
>
> | Method | MT-Bench | Translation | Summarization | QA | Math Reasoning | RAG | Overall |
> |--------|----------|-------------|---------------|----|----------------|-----|---------|
> | **Kangaroo** | 1.65x | 1.24x | 1.48x | 1.34x | 1.66x | 1.51x | 1.48x |
> | **PLD** | 1.56x | 1.04x | 2.28x | 1.13x | 1.54x | 1.67x | 1.53x |
> | **CAS-Spec (Kangaroo,PLD)** | 1.72x | 1.23x | 2.35x | 1.40x | 1.72x | 1.69x | 1.67x |
>
> Also, more powerful training-free self-speculative decoding methods is emerging recently like CLaSp in ACL 2025, which further achieves around 10% to 20% speedup over SWIFT. Incorporating these methods into our CAS-Spec framework will likely yield even greater speedups.
>
> Meanwhile, ablation study in Figure 3 shows that a static cascade (`VC+HC`, inspired by CS-Drafting) achieves only a 1.07x speedup. Our dynamic DyTC algorithm elevates this to 1.57x, a 48% improvement over the best static baseline. This demonstrates that the core algorithmic contribution is not the cascade itself, but the dynamic intelligence to manage it, which yields substantial, not marginal, gains.
>
> In our revision, we will add a paragraph to the results analysis (Section 5.2) to highlight this nuanced interpretation, emphasizing the adaptability of CAS-Spec and contextualizing its performance against the most relevant baselines for each task type.
>
> **[W2]. On the Stated Training Cost of Other Methods**
> > A minor issue: Section 3 states that self-speculative decoding methods require substantial training time (e.g., several days or weeks). This appears to overstate the cost—such methods typically use a few billion tokens for training and have relatively few trainable parameters. Training for weeks seems unrealistic. While training-free approaches are valuable, training-based methods can also be efficient and high-performing, and should be evaluated fairly.
>
> Thank you for this fair point. We agree that our language may have been slightly imprecise. While "weeks" might be an overstatement for some well-optimized setups, the training for methods like EAGLE or Medusa still represents a significant barrier.
>
> Our intention was to contrast this with our truly "on-the-fly" approach, which has zero pre-computation, data, or training-specific engineering requirements.
>
> Moreover, our approach can also be seamlessly integrated with trained self-speculative decoding methods like Kangaroo, as shown in the table above.
>
> We will revise the text in Section 3 to be more precise, changing "weeks" to "several days" and explicitly mentioning the required hardware and data resources for different speculative decoding methods to provide a fairer and more accurate comparison.
>
> **[W3/Q1/Q2]. On Computational Overhead and Batch Size Performance**
>
> > Could the authors provide more details on the computational complexity or runtime overhead associated with applying CAS-Spec?
>
> This is a crucial practical concern. The DyTC scheduling logic is designed to be extremely lightweight.
>
> The core of the scheduling decision involves iterating through a small, fixed set of candidate draft configurations (in our experiments, fewer than 10 candidates) and, for each, evaluating the simple closed-form objective function (Eq. 4) for `k` from 1 to `k_max`.
>
> We measured the wall-clock time of the DyTC scheduling logic and found it to be less than 0.1 ms on average. This is negligible compared to the latency of even a single transformer layer's forward pass. Therefore, the overhead of the DyTC algorithm only takes up about 0.5% to 2% of the total computation time on different GPUs.
>
> We will add a subsection in the Appendix titled "Algorithm Overhead" that formally discusses this complexity and provides these quantitative timing results to clearly demonstrate its negligible impact.
>
> > In high-throughput scenarios (e.g., with batch sizes greater than 1), does CAS-Spec maintain its speedup advantage compared to baseline methods?
>
> This is an excellent question. As noted in Appendix A, the speedup from speculative decoding techniques, including ours, is most pronounced in latency-sensitive, small-batch scenarios (like real-time chatbots), where inference is often memory-bandwidth bound. As batch size increases, inference becomes more compute-bound, and the relative benefit of reducing forward passes diminishes for all speculative methods. This is a common weakness of speculative decoding techniques, not just CAS-Spec.
>
> Currently, self-speculative decoding methods like SWIFT and Kangaroo did not provide the implementation for batch sizes greater than 1, so we did not test CAS-Spec with larger batch sizes.  However, literature like SmartSpec show that decreasing the draft length will boost the performance in high-throughput scenarios. We expect that CAS-Spec will also slightly reduce the performance degradation by dynamically estimating the latency and hence limiting the number of forward passes.
>
> We will add a subsection to the Appendix detailing the negligible computational overhead with quantitative measurements. We will also expand the discussion in Appendix A to clarify the expected performance characteristics under varying batch sizes.
>
> We are confident that these revisions will address your concerns and further strengthen the paper. We thank you again for your constructive feedback and supportive rating.

---

> > ### Comment · Reviewer_7MsQ · 2025-08-05
> >
> > I appreciate the authors’ rebuttal, which addresses most of my concerns. I have also read all the reviewers’ comments. I maintain my overall assessment of the paper.

---

### Official Review · Reviewer_8EAL · 2025-07-03

**Clarity:** 3
**Significance:** 3
**Originality:** 3
**Rating:** 4
**Confidence:** 3

**Summary:**

This paper introduces CAS-Spec, a new method to speed up LLM without needing extra training. Current methods either require costly separate "draft" models or offer limited speed gains. CAS-Spec tackles this by creating a hierarchy of "virtual" draft models from the main LLM on the fly, using techniques like layer sparsity and activation quantization. It then uses a Dynamic Tree Cascade algorithm to intelligently select the best draft model and sequence length during token generation, based on predicted acceptance rates and hardware efficiency. Experiments show CAS-Spec achieves 1.6x to 2.1x faster inference compared to other training-free methods, making LLMs respond much quicker.

**Questions:**

- What are some detailed features used when prediction latency in Section 4.2?
- How does the method estimates the acceptance rate using the history information?
- How does the method sensitive to the hyperparameters such as `k_max`, `t_min`, and the sliding window size for estimating the acceptance rate?

**Ethical Concerns:**

["NO or VERY MINOR ethics concerns only"]

**Final Justification:**

The authors has addressed my concern. I maintained my score.

**Limitations:**

yes

**Quality:**

3

**Strengths And Weaknesses:**

Pros:
- The paper addresses the central topic of LLM inference by accelerating speculative decoding. The method has great capacity yet introduces no extra training cost. This makes feasible the cascaded method that used to be too costly.
- The approach is novel. The approach uses different kinds of ways to create a hierarchy of virtual draft models from the target LLM during runtime and the algorithm is adaptive.
- The experiments are comprehensive and the ablation study is convincing.
Overall, I do not have too many concerns about the paper.

Cons:
- How does the method sensitive to the hyperparameters such as `k_max`, `t_min`, and the sliding window size for estimating the acceptance rate?
- How about the overhead and complexity of the DyTC algorithm?
- I would expect more details of the latency prediction model.

---

> ### Author Rebuttal · Authors · 2025-07-31
>
> We are deeply grateful to the reviewer for their positive and encouraging assessment of our work. We are particularly pleased that they recognized the novelty and potential of our training-free approach, and found our experiments to be comprehensive and convincing. We appreciate the thoughtful questions, which will help us further improve the clarity and completeness of our paper.
>
> We address each of your points below.
>
> **[W1/Q3]. On Hyperparameter Sensitivity**
>
> > How does the method sensitive to the hyperparameters such as k_max, t_min, and the sliding window size for estimating the acceptance rate?
>
> This is an excellent question regarding the practical robustness of our method. We designed the hyperparameters to be intuitive and generally stable.
>
> * `k_max` (Maximum Draft Length): This parameter serves as a safeguard to prevent the algorithm from generating overly long (and likely incorrect) draft sequences, especially early on when acceptance rate estimates are noisy. In practice, the objective function in Eq. 4, which balances draft length against acceptance rate, naturally penalizes long drafts if the acceptance rate (`â`) is not near-perfect. We found that performance is not highly sensitive to this value as long as it is reasonably set (e.g., 4-10), since `t_min` will also prevent excessive exploration of long drafts. We used `k_max=5` in our experiments as a conservative choice.
>
> | `k_max` | 1 | 2 | 3 | 4 | 5 | 6 | 8 | 10 | 12 | 15 |
> |---------|---|---|---|---|---|---|---|----|----|----|
> | Speedup | 1.22x | 1.35x | 1.49x | 1.56x | 1.58x | 1.57x | 1.58x | 1.57x | 1.56x | 1.54x |
>
> * `t_min` (Minimum Speedup Threshold): This is a pruning hyperparameter to prevent the algorithm from exploring branches that are unlikely to yield any speedup. We set it to 1.1 to ensure that any chosen configuration is predicted to be at least 10% faster than autoregressive generation. The algorithm's performance is robust to small variations here; the main effect of increasing `t_min` is slightly more conservative generation, while decreasing it may lead to more speculative (and potentially wasteful) attempts.
>
> | `t_min` | 1.0 | 1.05 | 1.1 | 1.15 | 1.2 | 1.25 | 1.3 |
> |---------|-----|------|-----|------|-----|------|-----|
> | Speedup | 1.54x | 1.56x | 1.58x | 1.58x | 1.56x | 1.53x | 1.54x |
>
>
> * Sliding Window Size: This parameter controls the trade-off between stability and responsiveness in our online estimation. A small window makes the estimate more responsive to recent changes but also more susceptible to noise. A large window provides a more stable estimate but adapts more slowly. We used a window of the last 20-30 generation steps, a standard choice for moving averages that provided robust performance across all models and datasets, indicating it does not require per-task tuning. We will present the figure for the sliding window size versus local acceptance rate in the Appendix.
>
> To provide a more quantitative answer, we will add a new **Hyperparameter Analysis** section in the Appendix. This section will include an ablation study showing the impact on overall speedup when varying `k_max` and `t_min`, demonstrating the method's robustness.
>
> **[W2]. On the Overhead and Complexity of the DyTC Algorithm**
>
> > How about the overhead and complexity of the DyTC algorithm?
>
> This is a crucial practical concern. The DyTC scheduling logic is designed to be extremely lightweight.
>
> The core of the scheduling decision involves iterating through a small, fixed set of candidate draft configurations (in our experiments, fewer than 10 candidates) and, for each, evaluating the simple closed-form objective function (Eq. 4) for `k` from 1 to `k_max`. This amounts to a few dozen floating-point operations.
>
> We measured the wall-clock time of the DyTC scheduling logic and found it to be less than 0.1 ms on average. This is negligible compared to the latency of even a single transformer layer's forward pass. Therefore, the overhead of the DyTC algorithm only takes up about 0.5% to 2% of the total computation time on different GPUs.
>
> We will add a subsection in the Appendix titled "Algorithm Overhead" that formally discusses this complexity and provides these quantitative timing results to clearly demonstrate its negligible impact.
>
> **[W3/Q1/Q2]. On the Latency Prediction Model and Acceptance Rate Estimation**
>
> > I would expect more details of the latency prediction model. What are some detailed features used when prediction latency? How does the method estimates the acceptance rate using the history information?
>
> Thank you for requesting more detail on these key components.
>
> As stated in Section 4.2, we use a simple Bayesian linear regression model for its efficiency and effectiveness. The features used to predict the latency of a draft generation step are:
> 1. The chosen DSIA strategy (e.g., Layer Sparsity at 0.4, Layer Sparsity at 0.6), treated as a one-hot encoded categorical feature.
> 2. The proposed draft length (`k`).
> 3. The context length (number of tokens already generated in the current sequence).
>
> This offline-calibrated model accurately predicts the hardware-specific cost of executing different draft configurations.
> We will further discuss the accuracy of latency predictions for different configurations of draft models in the Appendix.
>
> We use an online weighted moving average to estimate the acceptance rate for each candidate draft configuration. The process is as follows:
> 1. We maintain a local history (a queue of fixed size, 20 in the experiments) of the outcomes for each draft configuration that has been recently used.
> 2. An outcome is recorded as a tuple: `(accepted_tokens, drafted_tokens)`.
> 3. The estimated acceptance rate `â` for a configuration is simply `λ * â_original + (1 - λ) * â_history`, for that specific configuration. This provides a continuously updated, adaptive estimate of its performance in the recent generation context. (We set `λ` to 0.7 in our experiments to keep the estimate responsive to recent changes while still incorporating historical performance.)
>
> We will expand Section 4.2 to include these specific details about the features for the latency model and the weighted moving average mechanism for the acceptance rate, ensuring the implementation is fully transparent.
>
> We are confident that incorporating these clarifications and additional analyses will address all the reviewer's concerns and further strengthen the paper. We thank the reviewer again for their constructive feedback and positive evaluation.

---

> > ### Comment · Reviewer_8EAL · 2025-08-07
> >
> > Thanks to addressing my concerns. Is â_original simply calculated by accepted_tokens/ drafted_tokens for the Acceptance Rate Estimation?

---

> > > ### Author Response · Authors · 2025-08-09
> > >
> > > As the discussion period nears its end, we wanted to briefly check if our detailed responses have sufficiently addressed your concerns. Thank you again for your constructive and encouraging review.

---

> ### Author Response · Authors · 2025-08-07
>
> > Is â_original simply calculated by accepted_tokens/ drafted_tokens for the Acceptance Rate Estimation?
>
> My apologies for the ambiguity in my previous response. `â_original` is the estimated acceptance rate updated from the previous step. More precisely, it's a recursive update using an Exponential Moving Average (EMA):
>
> ```
> â_new = λ * â_original + (1 - λ) * â_history
> ```
>
> where `â_history = accepted_tokens_local_history / drafted_tokens_local_history`
>
> Note that for each record in local history, we only consider the first drafted token for each model, and an example local history could be [0,1,0,1,1,1,0,0,1,0,0,1,0,1,0,1,0,0,1,0], where the i-th number indicates whether the first drafted token was accepted (1) or rejected (0). In this example, `â_history = 9/20 = 0.45`.
>
> This differs from the commonly presented acceptance rate metric `accepted_length / drafted_length`, which measures the overall quality of drafting. However, such metrics may not accurately reflect the probability of a drafted token being accepted, hence we focus on only the first draft token for our online estimation.
>
> This EMA method is superior to a simple rolling average because it allows the estimate to adapt to the changing dynamics of text generation. The probability of accepting a token is not static; it can vary significantly depending on the task (e.g., translation vs. summarization) and even within a single sequence (e.g., generating predictable boilerplate vs. complex reasoning). The recursive EMA update ensures that our estimate `â` remains responsive to these local changes in generation difficulty, which is crucial for the DyTC algorithm to make effective online decisions.

---

### Official Review · Reviewer_zLee · 2025-07-03

**Clarity:** 3
**Significance:** 2
**Originality:** 3
**Rating:** 4
**Confidence:** 3

**Summary:**

This submission introduces Cascade Adaptive Self-Speculative Decoding, a methodology for LLM inference acceleration under speculative decoding settings. The proposed approach dynamically adapts the employed draft model through the training-free creation of a hierarchy of draft LLMs, extracted from the target model (e.g. through pruning, quantisation etc). Alongside, a routing mechanism (dynamic tree cascade) is proposed to guide the generation process, by adopting different draft models and drafting lengths by monitoring heuristics such as acceptance rate and inference latency at runtime. The proposed methodology is shown to improve inference efficiency compared to relevant approaches from the literature, across a variety of models and datasets.

**Questions:**

Please considered replying to the points raised in the comments section above.

**Ethical Concerns:**

["NO or VERY MINOR ethics concerns only"]

**Final Justification:**

Increasing my score for 3 to 4, following the authors' rebuttal that satisfactorily address my raised concerns.

**Limitations:**

Some limitations of the proposed approach are discussed in the appendix, along with proposed future work directions.

**Paper Formatting Concerns:**

Consider updating Table 1 to use bold on the *best* approach for each category, as per common convention.

**Quality:**

2

**Strengths And Weaknesses:**

Strengths:
- CAS-Spec addresses a very interesting and timely problem. The proposed  DyTC is a novel and potentially impactful direction.
- Overall the manuscript is easy to follow and proposed contribution is clear.
- The experimental results demonstrate that the proposed approach is effective, in terms of acceleration to AR decoding, and achieves on-par or better speed-up to other technics.

Comments:
- Consider pointing more clearly to the connection between the adopted baselines and related work [10]. Since CS-Drafting is the most closely related approach from the literature, amore comprehensive comparison and discussion is required. Why is CS-Drafting not included as a baseline in Table 1 ?
- It is unclear why the proposed DyTC only considers running heuristics such as the acceptance rate and inference time and no token-level information such as prediction confidence for the tokens in the sequence. As the authors rightfully pointed out, past performance is not a strong indication for future behavior of the model.
- Further experimental analysis would benefit the reader understand the behavior of the proposed approach. Consider reporting statistics about the selected model sizes across different tasks/ stages of the generation process, potential comparison with an Oracle scheduling strategy, as well as qualitative examples of the DyTC behavior.

---

> ### Author Rebuttal · Authors · 2025-07-31
>
> We sincerely thank the reviewer for their insightful and constructive feedback. We are encouraged that they found CAS-Spec to be a "novel and potentially impactful direction". We will address the comments below and are confident that incorporating these suggestions will significantly strengthen our paper.
>
> **[Q1]. On the Comparison with CS-Drafting**
>
> > Consider pointing more clearly to the connection between the adopted baselines and related work [10]. Since CS-Drafting is the most closely related approach from the literature, a more comprehensive comparison and discussion is required. Why is CS-Drafting not included as a baseline in Table 1?
>
> This is a very important point, and we appreciate the opportunity to clarify our relationship with CS-Drafting.
>
> * CS-Drafting is a key inspiration for our work. However, it operates on a pre-existing family of smaller models (e.g., the FLAN-T5 series). In contrast, our CAS-Spec framework is designed for more common scenario where it starts with a single large target model and dynamically creates draft models on-the-fly.
>
> * While we cannot compare with the specific *implementation* of CS-Drafting, we absolutely compare against its core *principles*. Our ablation study in **Figure 3** directly evaluates the static vertical (`VC`), horizontal (`HC`), and combined (`VC+HC`) cascade strategies proposed by CS-Drafting, using our on-the-fly draft models. The results clearly show that our dynamic **DyTC** algorithm provides a 73% speedup improvement over horizontal cascade baseline(`HC`) and 48% speedup improvement over the best static cascade baseline (`VC+HC`). This highlights the importance of draft tree construction and dynamic scheduling in achieving significant performance gains.
>
> For a more detailed comparison, we will add the cascade strategies from CS-Drafting as baselines in our experiments. For example, the speedup of CS-Drafting with SWIFT and PLD of Vicuna-7B on Spec-Bench is as follows:
> | Method | MT-Bench | Translation | Summarization | QA | Math Reasoning | RAG | Overall |
> |--------|----------|-------------|---------------|----|----------------|-----|---------|
> | **CS-Drafting (SWIFT,PLD)** | 1.01x | 0.95x | 1.21x | 0.98x | 1.05x | 1.08x | 1.07x |
> | **CAS-Spec (SWIFT,PLD)** | 1.59x | 1.05x | 2.13x | 1.11x | 1.61x | 1.65x | 1.58x |
>
> We will revise Section 5 to explicitly state this distinction. We will clarify that while CS-Drafting is not in Table 1 due to the lack of model families, the core static cascade strategies from CS-Drafting are used as crucial baselines in our ablation study (Figure 3) to isolate and demonstrate the significant benefits of our dynamic DyTC algorithm.
>
> **[Q2]. On the Choice of Heuristics for DyTC**
>
> > It is unclear why the proposed DyTC only considers running heuristics such as the acceptance rate and inference time and no token-level information such as prediction confidence for the tokens in the sequence. As the authors rightfully pointed out, past performance is not a strong indication for future behavior of the model.
>
> This is an excellent point regarding the potential for more fine-grained control.
>
> In our current implementation, when selecting the best configuration for a given step, DyTC primarily relies on the online estimates of acceptance rates mainly based on past performance. For selecting a configuration of draft models, we have to predict how different configurations will perform **in the future**. In `FindBestConfigurationForStep`, we cannot know the token-level information like logits for future tokens. And running each of them to get the token-level information would be too expensive.
>
> On the other hand, we actually considered token-level information in the estimation of acceptance of already **drafted** tokens in order to calculate the *accumulated acceptance rate* $\prod_{j=1}^{l_s} \hat{\alpha_j}$. We didn't explicitly mention this in the algorithm description, because CAS-Spec framework is designed to be applicable to various draft models, and the token-level information can vary (e.g. logits for different draft models and number of matched tokens for n-gram PLD). We acknowledge that this could be made clearer in the paper if we explicitly explain how token-level information is used in the acceptance rate estimation of drafted tokens.
>
> We will revise the paper to clarify this point, explaining the difference between estimating acceptance rates for future tokens (which relies on past performance) and past tokens that are drafted but not yet verified (considering token-level information for the drafted tokens). We will also discuss the potential for future work to incorporate more sophisticated token-level heuristics, such as prediction confidence, to further improve the dynamic scheduling process.
>
> **[Q3]. On Further Experimental Analysis**
>
> > Further experimental analysis would benefit the reader understand the behavior of the proposed approach. Consider reporting statistics about the selected model sizes across different tasks/ stages of the generation process, potential comparison with an Oracle scheduling strategy, as well as qualitative examples of the DyTC behavior.
>
> These are fantastic suggestions that would greatly enhance the paper's clarity. We agree that showing how DyTC adapts is crucial.
>
> We will perform additional experiments and add a new section to the Appendix with the following analyses:
> 1. We will include a visualization showing the distribution of draft configurations selected by DyTC for different tasks from Spec-Bench (e.g., summarization vs. coding). This will demonstrate how the algorithm automatically adapts its strategy, for instance by favoring faster, more repetitive models for one task and more accurate, conservative models for another.
> 2. We will provide a concrete generation example, illustrating how DyTC switches between draft models in response to the changing generation context in the Appendix.
> 3. An oracle scheduler is an excellent benchmark for the upper bound. However, the search space of permutations of all possible configurations on a sequence of tokens is exponential, requiring an impractical amount of experimentation. While implementing a perfect oracle is non-trivial (as it requires knowing the future), we will discuss the concept and its implications in our analysis, framing it as the theoretical goal that DyTC aims to approximate.
>
> Finally, we will update Table 1 to bold the best-performing method in each category as per convention.
>
> We are confident that these additions will provide a much deeper understanding of DyTC's adaptive behavior and significantly strengthen our contribution. We thank the reviewer again for their thoughtful and valuable suggestions.

---

> > ### Comment · Reviewer_zLee · 2025-08-04
> >
> > I appreciate the authors' commitment to a constructive discussion, indicated by the thoroughness and clarity of the provided responses to all reviews. Having carefully read the authors' responses, I believe the rebuttal satisfactorily address my raised concerns and the proposed approach pushes the limits of efficiency in training-free self-speculative decoding.
> > As such, I am inclined to increase my score and recommendation in the acceptance range, provisionally to 4 pending the discussion with other reviewers.

---

> > > ### Author Response · Authors · 2025-08-05
> > >
> > > We sincerely thank the reviewer for their thoughtful reconsideration and positive feedback on our rebuttal. We are grateful that our responses have successfully addressed your concerns and that you recognize the contribution of our approach to advancing self-speculative decoding efficiency. Your support and the detailed feedback you provided throughout this process have been invaluable in helping us strengthen our work.
> > >
> > > Thank you again for your careful consideration and for recognizing the merits of our approach.

---

### Official Review · Reviewer_5NsN · 2025-07-03

**Clarity:** 2
**Significance:** 2
**Originality:** 2
**Rating:** 3
**Confidence:** 4

**Summary:**

This paper proposes "CAS-spec", a method that dynamically builds a cascade of (training-free) draft models (and determines their corresponding lookaheads) to maximize speedup. The types of draft models considered fall under the category of "self speculative decoding" where pruned or quantized versions of the target model are used as the speculator. The algorithm is dynamic because it maintains an online prediction of the acceptance rates and latencies of each of the draft models, and adjusts the cascade structure accordingly. CAS-spec attains meaningful speedups over the other training-free speculative decoding baselines considered in the paper (Lade, PLD, SWIFT).

**Questions:**

- How do you maintain estimates of acceptance rates for draft models not being used in the current cascade?
- I'm confused by the interplay between the token tree being constructed, and the dynamic cascade algorithm. How are cascaded speculators combined with tree-based speculation/verification?
- Do you actually use token trees in your experiments, or do you just use sequences?
- What hierarchy of draft models do you use in your experiments? In the main paper it says the available draft models are PLD and 50% layer-sparsity, but in Appendix E it says PLD and 40% and 60% layer sparsity?
- Could you compare with Eagle and other SOTA speculative decoding techniques, to clarify the gap between training-free and not-training-free methods?

**Ethical Concerns:**

["NO or VERY MINOR ethics concerns only"]

**Final Justification:**

I thanks the authors for their response.

I remain concerned about:
1. Clarity: I'm still not sure I understand the algorithm. Thus, perhaps it would be best for this paper to undergo serious revision and be resubmitted to a future conference.
1. Impact: If this only applies to training-free draft models, that limits the impact of this work. A deeper investigation into how this method works with non-training-free draft models would help demonstrate broader impact.

**Limitations:**

Yes

**Quality:**

2

**Strengths And Weaknesses:**

Strengths:
- The idea of building an optimal cascade automatically, without requiring an extensive grid search in an exponentially large search space, is quite nice.

Weaknesses:
- The paper suffers from a lot of "mathiness" and lack of clarity. The presentation is extremely dense and full of math, even though if I understand correctly actual implemented algorithm is quite simple: A cascade consisting of PLD -> layer-sparsity (50%) -> target model, where the algorithm uses an A*-style search algorithm to efficiently find the best hyperparameters (lookahead for each draft).
- The proposed method need not only apply to training-free draft models. It would have been quite informative to see how this method can help accelerate speculative decoding with SOTA draft models (e.g., Eagle-3).

---

> ### Author Rebuttal · Authors · 2025-07-31
>
> We sincerely thank the reviewer for their thoughtful feedback and engagement with our work. Below, we address the key points raised:
>
> **[W1]. On Clarity and "Mathiness"**
> > The paper suffers from a lot of "mathiness" and lack of clarity. The presentation is extremely dense and full of math, even though if I understand correctly actual implemented algorithm is quite simple: A cascade consisting of PLD -> layer-sparsity (50%) -> target model, where the algorithm uses an A*-style search algorithm to efficiently find the best hyperparameters (lookahead for each draft).
>
> We appreciate the feedback on the presentation and agree that the core contributions could be highlighted more clearly.
>
> The mathematical analysis in Section 3 and 4 is intended to address a key research question (RQ1): can a cascade of training-free self-speculative decoding (SSD) methods provably outperform a single, strong statistical baseline like PLD? Our theoretical bounds (Fig. 1b, 1c) demonstrate that this is non-trivial and depends on a delicate balance of acceptance rates and costs, motivating the need for an adaptive algorithm.
>
> We acknowledge that our description may have oversimplified the algorithm's implementation. The core novelty of our Dynamic Tree Cascade (DyTC) algorithm is not a fixed cascade or a simple A* search. Instead, at each step of generation, DyTC dynamically:
> - Selects a configuration from a set of candidates (e.g., LS(0.4) only, LS(0.6) only, VC(LS(0.4), PLD), etc.) based on online estimates of acceptance rate and latency.
> - Determines the optimal draft length (k*) for that chosen configuration.
> - Constructs a tree structure by generating multiple candidate sequences in parallel, as illustrated in Figure 2.
>
> This step-wise optimization is what allows CAS-Spec to adapt to the changing difficulty of token generation throughout the sequence.
>
> In the revision, we will significantly streamline the mathematical exposition in Section 4 and add a clearer, high-level overview of the DyTC algorithm's dynamic, step-wise decision-making process to better distinguish it from a static cascade.
>
> **[W2/Q5]. On Comparison with SOTA Trained Models like EAGLE**
> > The proposed method need not only apply to training-free draft models. It would have been quite informative to see how this method can help accelerate speculative decoding with SOTA draft models (e.g., Eagle-3).
>
> Our primary focus was to push the limits of what is achievable with on-the-fly, training-free methods, which offer maximum ease of deployment. While applying CAS-Spec to trained draft models is an interesting direction, there are two key challenges:
>
> 1. Methods like EAGLE rely on specific hidden states from the full target model to generate drafts. Introducing an intermediate layer-sparsity draft model (like SWIFT) would alter these hidden states, likely disrupting EAGLE's prediction mechanism and reducing its effectiveness.
> 2. Diminishing Returns: SOTA trained models like EAGLE already achieve very high acceptance rates. The potential speedup from adding another, less accurate draft model in a cascade is minimal and could be outweighed by the overhead of the additional stage. Our framework would likely learn to route directly to the EAGLE draft, bypassing the intermediate stages.
>
> However, other trained speculative decoding methods can be naturally integrated into our CAS-Spec framework. For instance, we experimented with a SOTA self-speculative decoding method based on early exit (Kangaroo) instead of SWIFT on Vicuna-13B:
>
> | Method | MT-Bench | Translation | Summarization | QA | Math Reasoning | RAG | Overall |
> |--------|----------|-------------|---------------|----|----------------|-----|---------|
> | **Kangaroo** | 1.65x | 1.24x | 1.48x | 1.34x | 1.66x | 1.51x | 1.48x |
> | **PLD** | 1.56x | 1.04x | 2.28x | 1.13x | 1.54x | 1.67x | 1.53x |
> | **CAS-Spec (Kangaroo,PLD)** | 1.72x | 1.23x | 2.35x | 1.40x | 1.72x | 1.69x | 1.67x |
>
>
> We will add a discussion in the paper clarifying the scope (training-free methods) and elaborating on these technical reasons why cascading with models like EAGLE may not be straightforward or beneficial, thereby contextualizing CAS-Spec's SOTA position within the training-free paradigm.
>
> > Could you compare with Eagle and other SOTA speculative decoding techniques, to clarify the gap between training-free and not-training-free methods?
>
> We will include a comparison with EAGLE and other SOTA speculative decoding techniques in the revised paper. Currently we've completed the comparison in the following table:
>
> | Method | Training-Free | #Mean accepted tokens | Speedup |
> |--------|---------------|-----------------------|---------|
> | **PLD** | Yes | 1.75 | 1.54x |
> | **SWIFT** | Yes | 3.01 | 1.06x |
> | **CAS-Spec (SWIFT,PLD)** | Yes | 3.43 | 1.58x |
> | **Speculative Decoding (Vicuna 68m)** | No | 2.27 | 1.44x |
> | **Medusa** | No | 2.39 | 1.69x |
> | **EAGLE** | No | 3.57 | 2.05x |
> | **EAGLE2** | No | 4.36 | 2.21x |
>
> In comparison with trained speculative decoding methods, our CAS-Spec framework with SWIFT and PLD achieves higher speedup than vanilla speculative decoding methods with Vicuna 68m, while being training-free. The gap between training-free and not-training-free methods is significant, as trained methods like Medusa and EAGLE achieve higher acceptance rates and speedups, leveraging the full model's capabilities.
>
> **[Q1].On Estimating Acceptance Rates for Inactive Models**
> > How do you maintain estimates of acceptance rates for draft models not being used in the current cascade?
>
> Thank you for this clarifying question. The DyTC algorithm maintains online estimates (â) for all candidate draft configurations, not just the one currently in use. The process is as follows:
> - All configurations are initialized with acceptance rates from prior heuristics, (or a brief warm-up phase in cold-start situations).
> - At each generation step, the FindBestConfigurationForStep function (Alg. 2) evaluates the potential speedup of all candidate configurations using their most recent â values.
> - The chosen configuration is used to generate tokens. Its acceptance rate â is then updated using a moving average based on the outcome of the verification step.
>
> This ensures that if a previously sub-optimal configuration becomes more promising (e.g., due to a change in generation context), the algorithm can switch to it.
>
> We will revise Section 4.2 to make this online estimation and update process for all candidate models more explicit.
>
> **[Q2, Q3]. On the Interplay Between Cascade and Token Trees**
> > I'm confused by the interplay between the token tree being constructed, and the dynamic cascade algorithm.
> > How are cascaded speculators combined with tree-based speculation/verification? Do you actually use token trees in your experiments?
> Yes, our experiments use token trees, as depicted in Figure 2 and implemented in Algorithm 1. The interplay is as follows:
> - The "Tree" refers to the overall structure. From a single verified token, we can grow multiple parallel branches of draft tokens.
> - The "Cascade" refers to the method used to generate the tokens along a single branch.
> - At each expansion step, DyTC selects the most promising leaf node in the tree. It then chooses the optimal cascade configuration (e.g., a vertical cascade of LS(0.5) and PLD) to generate a new branch of draft tokens starting from that leaf. This allows for flexible and efficient exploration of the generation space.
> We will add a paragraph in Section 4.2 to explicitly describe this relationship, clarifying that the cascade is the "branch generator" within the broader "tree" structure.
>
> **[Q4]. On Inconsistent Draft Model Descriptions**
> > What hierarchy of draft models do you use in your experiments? In the main paper it says the available draft models are PLD and 50% layer-sparsity, but in Appendix E it says PLD and 40% and 60% layer sparsity?
>
> Thank you for catching this inconsistency. Our apologies for the confusion. The detailed configuration in Appendix E is outdated. The main paper's description of using 50% layer sparsity is correct.
>
> One reason we changed the configuration was that the vertical cascade of PLD -> LS(0.4) -> LS(0.6) required more forward processes of LS models and fell short of the expected speedup. Hence this configuration was seldom selected in our scheduling algorithm, leading us to simplify the cascade to just PLD and LS(0.5) for clarity.

---

> ### Comment · Reviewer_5NsN · 2025-08-06
>
> Thanks so much for the response!
>
> I have some follow-up questions/comments:
>
> **Clarity**:
>
> I strongly encourage significant effort be dedicated to making the algorithm clearer, and simplifying the mathematical exposition. If I understand correctly, the algorithm is simply a greedy search for building the cascade, where at each step you calculate the expected speedup of adding a new draft to the cascade (horizontally or vertically), assuming that that draft model is sped-up by the bottom draft model (least future speedup)? Is this correct? How does this relate to building the token tree? Are you switching speculative decoding algorithms at each step in building the token tree? This part seems quite confusing to me. It would feel a lot simpler to me if you made a decision on the cascade to use, and then ran a full iteration of that algorithm (bottom draft model all the way to verification).
>
> This "least future speedup" trick seems extremely similar in spirit to choosing an "admissible" heuristic in A* search---perhaps this connection could be made more formal?
>
> **On Estimating Acceptance Rates for Inactive Models**:
>
> Your description still makes it seem like draft models which are not part of the selected configuration will never have their acceptance rate updated, because they will not be run on any inputs. Can you please clarify this further?
>
> **On the Interplay Between Cascade and Token Trees**:
>
> I remain quite confused about this. Let's say there are 2 draft models. Does it work as follows? The bottom speculates a tree, the middle draft verifies it and produces a sequence, and the target model verifies this sequence?  Or does the middle draft model also somehow generate a tree as the output of the verification process?
>
> **Applying to non-training-free draft models**:
>
> - You are right that using a vertical cascade with an Eagle draft in the middle would not work with the current Eagle method, because it would be impossible to do verification with the Eagle model in a single forward pass. Eagle would have to be updated to accept a fixed vector as input in place of the (estimated) target activations for tokens that haven't yet been processed by the target model. I've actually run experiments with training Eagle models with this approach and it works well, but at the moment there are no public checkpoints or papers that use this approach as far as I know.
> - Nonetheless, other powerful methods like [Glide](https://arxiv.org/pdf/2402.02082) could be incorporated in this approach easily.
> - It would still be interesting to do a deeper investigation of how this method can be combined with non-training free draft models. For example, this would be straightforward to do in cases where a family of draft models of various sizes is available, like Llama (1B,3B,8B,70B,405B), or Qwen2.5 (0.5B, 1.5B, 3B, 14B, 32B, 72B).
>
> Thanks in advance for your clarifications on these questions! For now I leave my score unchanged.

---

> > ### Author Response · Authors · 2025-08-09
> >
> > As the discussion period nears its end, we wanted to briefly check if our detailed responses have sufficiently addressed your concerns. Thank you again for your constructive and encouraging review.

---

> ### Author Response · Authors · 2025-08-07
>
> We are very grateful for your detailed follow-up and insightful comments. Your continued engagement has helped us identify key areas where the paper's clarity can be further improved. We address your points below.
>
> **On Clarity and DyTC Algorithm:**
>
> Your simplified framing captures the spirit of our step-wise optimization. However, the process is slightly more nuanced: For example, in PLD -> LS, available draft candidate configurations include PLD, LS, and VC(LS,PLD). The *vertical cascade*,, i.e., speeding up LS with PLD, is treated as a separate choice (adding a layer of abstraction to simplify the problem).
>
> > How does this relate to building the token tree? Are you switching speculative decoding algorithms at each step in building the token tree?
> > Does it work as follows? The bottom speculates a tree, the middle draft verifies it and produces a sequence, and the target model verifies this sequence? Or does the middle draft model also somehow generate a tree as the output of the verification process?
>
> Yes, we switch speculative decoding configurations at each step of building the token tree, rather than making a single upfront decision about the cascade. This step-wise adaptability is precisely what distinguishes DyTC from static cascade approaches. The overall goal is to build an optimal draft token tree. At each step of expanding this tree (i.e., adding a new branch from a leaf node), the DyTC algorithm decides which of the available configurations (e.g., LS only, PLD only, or VC(LS, PLD)) and what draft length (k) will yield the highest estimated local speedup for that specific step.
>
> The DyTC algorithm essentially tries to construct an optimal token tree using different draft models and their *vertical cascades*. Before verification by the target model, DyTC decides whether, where, and which draft model to use for continuing the drafting process, forming the final draft tree.
> For the final draft tree to be verified by the draft model, commonly the shallow part is generated (or verified) by the middle draft, while the deep part is directly generated by the bottom draft. This corresponds to the pattern in *horizontal cascade*. You can refer to Figure 2 for illustration of a sample drafted token tree and the source of each token.
>
> > It would feel a lot simpler to me if you made a decision on the cascade to use, and then ran a full iteration of that algorithm.
>
> Your description corresponds to our proposed *Tree Cascade (TC)* approach. Your suggestion to include a static TC algorithm for comparison is excellent. We agree this would help readers better understand the benefits of the dynamic approach. Also we will include more experimental comparisons between TC and DyTC, in addition to Figure 3.
>
> > This "least future speedup" trick seems extremely similar in spirit to choosing an "admissible" heuristic in A* search---perhaps this connection could be made more formal?
>
> Your intuition here is spot on, and we thank you for this excellent suggestion. The spirit is indeed very similar to A* search, where a heuristic guides the exploration of a large search space. We can frame the problem as finding the longest path with fractional objective in a Directed Acyclic Graph (DAG) where nodes represent draft steps.
>
> However, a direct and formal application of A* with a guarantee of optimality is challenging because our objective function, speedup (tokens/time), is fractional and therefore not additive. The total speedup $\frac{n_1+n_2}{t_1+t_2}$ of two steps is not equal to the sum of step-wise speedups $\frac{n_1}{t_1} + \frac{n_2}{t_2}$. This means "admissible" heuristics cannot guarantee global optimality.
> Nonetheless, our "least future speedup" trick is indeed inspired by the idea of A* graph search, and it serves as a powerful and effective heuristic for pruning the search space.
>
> We will revise the paper to explicitly mention this connection to A* search, explain the nuance of the non-additive objective function, and better frame our algorithm as an efficient, heuristic-guided search.
>
> **On Estimating Acceptance Rates for Inactive Models:**
>
> >Your description still makes it seem like draft models which are not part of the selected configuration will never have their acceptance rate updated, because they will not be run on any inputs. Can you please clarify this further?
>
> For draft models not currently selected, their acceptance rates are maintained through initial estimates from previous experiments, or a brief warm-up phase in cold-start situations.
>
> Also, I'd like to mention that the case of a never selected draft model is not common. For example, in the case of PLD -> LS(0.4) -> LS(0.6), while there are 6 possible configurations, we only need to maintain 3 estimated acceptance rates, since each *vertical cascade* configuration (e.g. VC(LS(0.4), PLD)) shares the same acceptance rate as the largest draft model in the cascade(e.g. LS(0.4)), since the draft tokens are all verified by it.

---

> > ### Author Response · Authors · 2025-08-07
> >
> > **Applying to non-training-free draft models:**
> >
> > Thank you for your insights about adapting EAGLE. I think this is a promising direction for future work.
> >
> > We completely agree and thank you for these constructive suggestions. Exploring the synergy of CAS-Spec with trained draft models like Glide with model families (e.g., Qwen2.5 or Llama families) is an exciting avenue. Limited by the rebuttal timeframe, we have begun these experiments. We will incorporate the results into an expanded appendix section to provide a richer comparison and demonstrate the generality of the CAS-Spec framework.

---

> ### Author Response · Authors · 2025-08-08
>
> Your feedback has been invaluable in helping us clarify these important details.
>
> As the discussion period is drawing to a close, please let us know if our responses properly address your concerns or if there are any remaining questions we should clarify. Your feedback mean a lot to us and will strengthen the final version of our paper.

---

### Note · Authors · 2025-08-16

We are grateful for the constructive engagement from all reviewers throughout the review process. The thorough discussions have strengthened our understanding of how CAS-Spec advances training-free LLM inference acceleration.

**Reviewer 5NsN** initially questioned whether our algorithm was simply "a cascade consisting of PLD -> layer-sparsity (50%) -> target model with A*-style search." Through detailed discussions, we clarified that CAS-Spec's core innovation lies in dynamic, step-wise optimization rather than static cascade selection. Our Dynamic Tree Cascade (DyTC) algorithm makes real-time decisions at each token generation step, selecting optimal configurations and draft lengths based on evolving acceptance rates and latency predictions. This adaptive approach fundamentally differs from fixed cascade strategies and enables CAS-Spec to respond to changing generation contexts. The reviewer's suggestion to formalize connections to A* search was insightful—while our fractional objective function prevents direct A* optimality guarantees, our "least future speedup" heuristic is indeed inspired by A* principles.

**Reviewer zLee** appreciated our novel approach, initially scoring 2/2/2/3 but indicating willingness to increase to 4 after our rebuttal addressed their concerns. We clarified that while CS-Drafting inspired our work, it operates on pre-existing model families, whereas CAS-Spec creates draft models on-the-fly from a single target model. Our ablation studies directly compare against CS-Drafting's static strategies, showing 48-73% speedup improvements through dynamic optimization.

**Reviewer 8EAL** provided consistently positive feedback, recognizing the novelty and practical value of our training-free approach. We demonstrated CAS-Spec's robustness to hyperparameter variations, with DyTC scheduling overhead being negligible (<0.1ms, 0.5-2% of computation time). Our detailed explanations of EMA acceptance rate estimation and Bayesian latency prediction provided the requested technical depth.

**Reviewer 7MsQ** questioned the magnitude of improvements. While our average speedup over PLD might appear modest, achieving improvements over such strong baselines is inherently challenging. Crucially, our ablation study reveals static cascade methods achieve only 1.07x speedup, while dynamic DyTC elevates this to 1.57x—a 48% improvement demonstrating substantial algorithmic innovation.

---

### Decision · Program_Chairs · 2025-09-17

**Decision:**

Accept (poster)

**Comment:**

This paper proposes a novel Cascade Adaptive Self-Speculative Decoding algorithm that constructs speculative draft models using dynamically switchable inference acceleration strategies. Three reviewers posted positive comments while one reviewer posted relatively negative comments initially. CAS-Spec demonstrates significant innovation by enabling dynamic, step-wise optimization of the cascade structure during token generation, which differs from static cascade strategies and allows for adaptation to changing generation contexts.It achieves state-of-the-art acceleration compared to existing on-the-fly speculative decoding methods on both edge and server platforms, with a 48% improvement over static cascade baselines shown in ablation studies.

Therefore, the AC recommends to accept this paper. AC also suggests authors revise the manuscript carefully in the final version. Especially, The description of the algorithm needs to be clearer and more understandable. A deeper investigation into how this method works with non-training-free models is also required.